# Advances in Diet and Physical Activity in Breast Cancer Prevention and Treatment

**DOI:** 10.3390/nu16142262

**Published:** 2024-07-13

**Authors:** Amr Khalifa, Ana Guijarro, Alessio Nencioni

**Affiliations:** 1Department of Internal Medicine and Medical Specialties, University of Genoa, Viale Benedetto XV 6, 16132 Genoa, Italy; anaguijarroa@yahoo.es; 2IRCCS Ospedale Policlinico San Martino, Largo Rosanna Benzi 10, 16132 Genoa, Italy

**Keywords:** Mediterranean diet, fasting, calorie restriction, ketogenic diets, vegan or plant-based diets, physical activity, lifestyle interventions, breast cancer

## Abstract

There is currently a growing interest in diets and physical activity patterns that may be beneficial in preventing and treating breast cancer (BC). Mounting evidence indicates that indeed, the so-called Mediterranean diet (MedDiet) and regular physical activity likely both help reduce the risk of developing BC. For those who have already received a BC diagnosis, these interventions may decrease the risk of tumor recurrence after treatment and improve quality of life. Studies also show the potential of other dietary interventions, including fasting or modified fasting, calorie restriction, ketogenic diets, and vegan or plant-based diets, to enhance the efficacy of BC therapies. In this review article, we discuss the biological rationale for utilizing these dietary interventions and physical activity in BC prevention and treatment. We highlight published and ongoing clinical studies that have applied these lifestyle interventions to BC patients. This review offers valuable insights into the potential application of these dietary interventions and physical activity as complimentary therapies in BC management.

## 1. Introduction

Studying the effect of a certain diet on a defined health problem is difficult in principle given that any diet is highly complex and entails the intake of many different types of nutrients with the ability to exert multiple simultaneous effects in the body. Yet recent and past studies have defined specific dietary patterns, particularly diets that are typically adopted in defined geographical areas, that lend themselves to this type of studies. The Mediterranean diet (MedDiet), which was first described by Ancel Keys some seventy years ago [1], is possibly the most notable example in this area and it is also the type of diet that has shown the most convincing cancer-preventing effects in humans. Thus, the antitumor effects of the MedDiet are going to be the main focus of this section. Yet we will also address the possible role of other dietary approaches, such as fasting or modified fasting (MF), calorie restriction (CR), the ketogenic diet (KD), and vegan or plant-based diets, in breast cancer (BC) treatment, although their use remains experimental and is not devoid of concerns (mostly in relation to their potential to cause malnutrition in patients who, due to their condition or to the therapies they are receiving, are already at risk for this type of complication).

The MedDiet is a plant-based diet composed of low saturated fat content, high monounsaturated fats (primarily from the high intake of olive oil), and abundant dietary fibers. It includes vegetables, fresh fruit, legumes, non-refined cereals, nuts, poultry, moderate amount of sea food and dairy products, low intake of red and processed meat, and moderate alcohol consumption [2]. First described between 1958 and 1970 by Keys and his research team [1], the MedDiet is based on the traditional foods and drinks of countries in the Mediterranean region, particularly those around the Mediterranean Sea. The MedDiet is rich in antioxidant compounds and bioactive elements with anti-inflammatory effects, and it has a low glycemic index [3,4]. Adherence to the MedDiet has been associated with decreased risk of several chronic diseases, such as cardiovascular disease [5], diabetes [6], neurodegenerative diseases [7], and cancer [8]. Epidemiological evidence has demonstrated increased longevity in healthy people [2] and reduced overall mortality [8] with higher adherence to the MedDiet. Further epidemiological studies strongly support the hypothesis that the main components of the MedDiet play a protective role against several types of cancers, especially those of digestive tract [9] and hormone-dependent cancers [10,11].

Physical activity plays a crucial role in both the management and prevention of BC, providing a multitude of benefits that enhance patient outcomes and reduce disease risk. Regular exercise can improve the efficacy of anticancer therapies by boosting overall fitness and resilience [12], which helps patients better tolerate treatments such as chemotherapy and radiation. Physical activity is also known to alleviate side effects associated with these therapies, including fatigue, nausea, and pain, thereby enhancing patients’ quality of life (QoL) [13]. Additionally, maintaining an active lifestyle helps combat obesity, a significant risk factor for BC. By regulating body weight, exercise reduces the likelihood of developing BC and supports a better prognosis for those already diagnosed. The multifaceted role of physical activity underscores its importance in comprehensive BC care strategies, promoting both prevention and effective management [14].

Here, we discuss the biological rationale for using the MedDiet and physical activity to prevent and treat BC. We also highlight published and ongoing clinical studies that have applied the MedDiet or physical activity to patients with BC. At the molecular level, we offer insight into the current understanding of how the MedDiet and physical activity impact BC progression. We also discuss the potential of experimental diets, such as periodic fasting or MF, CR, KD, and vegan or plant-based diets in BC prevention and treatment.

## 2. Exploring Potential Cancer-Fighting Mechanisms of the MedDiet

The precise mechanism through which a traditional MedDiet exerts its beneficial effects in combating heart disease, cancers, and other metabolic conditions is not fully understood. Various nutrient components present in the MedDiet have been hypothesized to contribute to the observed beneficial effects [3]. Figure 1 provides an overview of these beneficial effects.

### 2.1. Antioxidant and Anti-Inflammatory Effects

The positive correlation between the MedDiet and cancer prevention may be attributed to the antioxidant and anti-inflammatory effects of the bioactive substances found in its foods, such as legumes, fruits, nuts, vegetables, fish, and extra-virgin olive oil (EVOO). These effects play a crucial role in inhibiting the proliferation of cancer cells and preventing BC. The MedDiet is rich in antioxidant substances, including polyphenols, phytosterols, and other bioactive compounds like hydroxycinnamic derivatives, quercetin, resveratrol, oleuropein, and hydroxytyrosol, which appear to influence oxidative stress and DNA damage [15,16].

Dietary polyphenols found in EVOO, fruits, vegetables, whole grains, and nuts, which are abundant in the MedDiet, could be utilized for cancer prevention and treatment due to their antioxidant effects that protect against reactive oxygen species and reactive nitrogen species [17]. Fruits and vegetables are rich sources of various carotenoids, vitamins (such as vitamin C, vitamin E, and retinol), folates, and flavonoids, all of which are known for their antioxidant properties that help prevent DNA damage [18,19]. Lycopene, a natural antioxidant found in tomatoes, exhibits antitumor activity through its antioxidant, anti-inflammatory, and immunomodulatory properties. It regulates the expression of the antioxidant response element gene and competes with estrogen for estrogen receptor (ER)α and ERβ, reducing the transactivation of estrogen response elements in DNA [20]. Lycopene also inhibits the growth of BC cells by blocking NF-κB signaling [21]. Naturally occurring retinoids, such as all-trans-retinoic acid and 9-cis retinoic acid, are structurally and functionally similar to vitamin A and are found in meat, poultry, fish, and dairy products, which are also part of the MedDiet [2]. Their beneficial effects against cancer are due to their ability to modulate various cancer hallmarks, including cell cycle growth arrest, apoptosis, inflammation, and angiogenesis [22].

Olive oil is the primary source of fat in the MedDiet, and its consumption is associated with a reduced incidence of and mortality from cancer [23]. Olive oil is rich in monounsaturated fatty acids, particularly oleic acid, and contains over 200 minor compounds. Among these, several exhibit potent bioactivities, such as phenolic compounds (including phenolic alcohols like oleuropein, tyrosol, and hydroxytyrosol; secoiridoids such as oleuropein, oleocanthal, and oleacein; oleanolic acid, maslinic acid, and lignans like pinoresinol), flavonoids, triterpenes, and vitamin E. These compounds contribute to the anti-inflammatory and antioxidant properties associated with olive oil [24]. However, the specific constituents vary considerably based on the oil’s quality, including its physicochemical and organoleptic attributes.

A systematic review and meta-analysis of 19 observational studies, involving 13,800 patients and 23,340 controls and conducted between 1990 and 2011, found that olive oil intake is inversely related to cancer prevalence [25]. Besides, adherence to the MedDiet supplemented with EVOO in the Prevención con Dieta Mediterránea (PREDIMED) randomized trial resulted in a 62% reduction in invasive BC incidence compared to the control group [26]. This groundbreaking trial was the first to confirm the protective effects of EVOO within the MedDiet against BC.

However, not all cohort studies have demonstrated a correlation between olive oil intake and a reduced risk of BC. Specifically, two comprehensive prospective cohort studies conducted in the United States aimed to explore the relationship between olive oil consumption and BC risk within a non-Mediterranean population [27]. These studies found no significant correlation between olive oil intake and the risk of BC [27], in contrast to earlier findings from case–control studies conducted in traditional Mediterranean countries like Greece, Spain, Italy, and France [26,28,29]. These latter studies, conducted among Mediterranean populations, commonly suggested that consuming olive oil was associated with a decreased risk of BC.

### 2.2. Cholesterol Absorption and Biosynthesis

Fresh fruit, vegetables, legumes, and nuts present in the MedDiet inhibit cholesterol absorption and biosynthesis [30]. Results from a random subsample of individuals in the PREDIMED study (ISRCTN35739639) showed that the MedDiet enhances cholesterol efflux capacity by reducing the activity of cholesteryl ester transfer protein, thus improving the protective role of HDL in reverse cholesterol transport [31]. Recently, our group demonstrated that dietary regimens such as periodic fasting sensitize cancer to cholesterol biosynthesis inhibitors, resulting in decreased cholesterol levels within tumors. Fasting promoted cholesterol efflux in cancer cells to mature HDLs, and the administration of HDLs to pancreatic ductal adenocarcinoma (PADC) xenograft-bearing mice increased fasting’s effectiveness in reducing intratumor cholesterol and slowing tumor growth, highlighting the role of targeting cholesterol in cancer prevention [32].

Omega-3 fatty acids, abundantly found in seafood (especially sardines and mackerel, typical of the MedDiet) and in nuts (almonds, walnuts, and pumpkin seeds), help slow cancer development, lower the risk of BC, improve overall survival [33,34], and reduce cholesterol levels [35]. Studies have shown that a higher intake of red and processed meat correlates with an increased risk of cancer and other chronic diseases [36,37].

### 2.3. Autophagy Induction/Apoptotic and Antiproliferative Effects

Autophagy is a key pathway that facilitates the degradation and recycling of cellular components. The role of autophagy in cancer is complex and depends on the tumor stage, biology, and surrounding microenvironment. The nutrients in the MedDiet are rich in dietary polyphenols. Resveratrol, a polyphenolic compound found in nuts, red grapes, berries, and other MedDiet components, is a potent inducer of autophagy [38]. Autophagy has powerful antioxidant, anti-inflammatory, and apoptotic effects in BC cells through the activation of p53-dependent pathways [39]. Resveratrol also inhibits DNA methyltransferase and alters chromatin modification in BC [40]. Other polyphenols present in EVOO, such as oleocanthal and oleuropein, have also been reported to induce autophagy [41].

### 2.4. Gut Microbiota

It has been widely recognized that diet explicitly modulates the gut microbiome [42]. Several studies suggest that the MedDiet exerts a beneficial role in the gut microbiota composition [30,43,44]. High adherence to the MedDiet increases the levels of *Firmicutes* in the subjects’ gut microbiota [45]. Some reports suggest that the MedDiet promotes growth of short-chain fatty acids (SCFAs), producing *Bacteroidetes* and limiting the development of *Firmicutes* [46]. Research by De Filippis et al. [43] demonstrated that individuals who adhere more strictly to the MedDiet exhibit a higher percentage of SCFAs and fiber-degrading bacteria, such as *Prevotella*, in their feces. Conversely, subjects with poor adherence to the MedDiet show higher concentrations of urinary trimethylamine oxide, a potential risk factor for cardiovascular disease known to alter cholesterol and activate inflammatory pathways [NCT02118857].

The positive impact of the MedDiet on gut microbiota has not been extensively studied in the context of cancer treatment. However, in a four-arm randomized, controlled trial (RCT) [NCT04753359] involving 232 obese African Americans, participants were assigned to one of four interventions: a weight-stable MedDiet alone (Med-A) lifestyle intervention, calorie restriction for weight loss (WL-A) with no change in diet composition, a MedDiet weight loss (WL-Med) lifestyle intervention, or a control group. This RCT was designed to investigate the effects of these interventions on the concentration and composition of circulating and fecal bile acids, gut microbiota, metabolic function, and gene expression profiles of exfoliated intestinal epithelial cells. The investigators hypothesized that targeting the bile acids–gut microbiome axis to suppress the abundance, growth, and metabolic activity of hydrogen sulfide and the tumor promoter deoxycholic acid-producing bacteria through diet and weight loss (WL) may reduce colorectal cancer risk, particularly among African Americans [NCT04753359].

Polyunsaturated Fatty Acids (PUFAs), particularly omega-3, are known to reduce inflammation [47,48], which may contribute to preventing heart disease, cancer, and various other chronic diseases. Additionally, a high-fat diet rich in omega-3 from fish oil has been shown to induce significant reactive oxygen species and promote protumor macrophage death, thus preventing the development of BC [49]. The anti-inflammatory effects induced by PUFA administration have been found to be mediated through reductions in *Firmicutes* and *Blautia* in gut microbiota [47,50], as well as by improving the intestinal epithelial barrier, thereby reducing its permeability in colitis [47].

### 2.5. Clinical Evidence of the Beneficial Role of the MedDiet in BC Patients

The association between the MedDiet and BC is evident in postmenopausal females, while the results for premenopausal females remain inconsistent [4]. In a case–control study, Castelló and colleagues confirmed the detrimental effect of a Western diet on BC risk, whereas the MedDiet demonstrated a protective role against BC mortality risk, particularly among Spanish females with triple-negative BC [51]. Similarly, the MCC-Spain multicase–control study analyzed three dietary regimens—Western diet (high intakes of fatty and sugary products and red and processed meat), Prudent diet (high intakes of low-fat dairy products, vegetables, fruits, whole grains, and juices), and the MedDiet. The results confirmed the associations of the Western diet with BC risk in both premenopausal and postmenopausal females. While the Prudent diet showed no effect on BC, the MedDiet appeared protective among postmenopausal females [52].

In a prospective cohort study, Buckland et al. evaluated the association between adherence to the MedDiet and BC risk among 335,062 women recruited from ten European countries between 1992 and 2000 and followed for an average of 11 years. Higher adherence to the MedDiet was associated with a lower risk of BC in postmenopausal females, particularly in ER−/progesterone receptor (PR)− tumors, whereas this correlation was not observed in premenopausal females [23]. In another prospective cohort study involving 13,270 BC survivors, researchers found that consuming a MedDiet before BC diagnosis may improve long-term prognosis, especially after menopause. The MedDiet exerted a stronger protective effect in cases of metastatic BC patients [53].

In the PREDIMED randomized, multicenter, controlled field trial conducted in Spain [ISRCTN35739639], 35 participants were identified as confirmed incident cases of malignant BC. Patients allocated to the MedDiet supplemented with EVOO demonstrated a 62% relative lower risk of malignant BC compared to those allocated to the control diet (advice to reduce dietary fat). However, participants in the MedDiet supplemented with nuts showed a nonsignificant risk reduction compared with women in the control group. When both MedDiet groups were combined, a 51% relative risk reduction was observed [26].

In a recent population-based case–control study comprising 818 BC patients and 935 healthy controls, the adherence to a migrated Chinese version of the MedDiet, referred to as the vegetable-fruit-soy dietary pattern, was assessed using a modified version of the MedDiet score called the alternate Chinese Diet Score. In this diet, soy foods, rapeseed oil, and coarse cereals replaced legumes, olive oil, and whole grains, reflecting the cuisine of the region. The study examined the association between adherence to the vegetable-fruit-soy diet and BC risk, stratified by menopause status (pre- or postmenopausal) and receptor status [ER, PR status, and human epidermal growth factor 2 (HER2) oncogene expression]. The results suggested that the vegetable-fruit-soy dietary pattern was inversely associated with postmenopausal BC risk, with a stronger association observed among ER− and ER−/PR− subtypes, while no significant association was observed with ER+, PR+, or ER+/PR+ subtypes [54].

Finally, we would like to draw the reader’s attention to the fact that clinical research on the efficacy of the MedDiet for BC patients has yielded mixed findings. While some studies suggest that adherence to the MedDiet is associated with a reduced risk of BC incidence and recurrence [26,28,29], other research has not consistently supported these findings [27]. Several observational studies have indicated a potential protective effect of the MedDiet against BC. These studies have reported lower BC incidence rates among individuals adhering to a Mediterranean dietary pattern characterized by high consumption of fruits, vegetables, whole grains, fish, and olive oil. Additionally, adherence to the MedDiet has been linked to favorable changes in biomarkers associated with BC risk, such as lower levels of inflammation and oxidative stress. However, findings from clinical trials investigating the direct impact of the MedDiet on BC outcomes have been less conclusive. Some trials have failed to demonstrate a significant reduction in BC risk or improvement in survival outcomes among participants assigned to a Mediterranean dietary intervention compared to control groups. These trials often face challenges such as participant adherence to the prescribed dietary regimen, variation in study design and duration, and difficulty in isolating the effects of diet from other lifestyle factors. Table 1 summarizes the clinical studies investigating the effects of the MedDiet in BC patients.

## 3. Potential of Periodic Fasting or MF in BC Treatment

As anticipated, mounting evidence also suggests the potential usefulness of regimens of short-term “extreme” dietary restriction, i.e., fasting or MF, in BC treatment. MF typically consists of a vegan, very low-calorie (e.g., 300–1000 kcal/day), low-protein, and low-sugar diet lasting a few days (e.g., 3–5 days), which is repeated every few weeks (in humans) or weekly (in mice) and is meant to generate the physiological effects of water-only fasting without actually imposing complete abstinence from food [55].

Early studies indicated the ability of fasting or MF to make chemotherapeutics such as doxorubicin or cyclophosphamide more active in mouse BC models [56,57]. This effect was ascribed to fasting’s ability to lower the growth factor, insulin-like growth factor 1 (IGF1), as well as to the reduction in tumor HO-1 levels which is observed during starvation, with HO-1 reduction being responsible for blunting Tregs-dependent obstruction of antitumor immunity [57].

Studies by our group also highlight the potential of fasting or MF regimens to synergize with endocrine therapy (ET) for hormone receptor positive (HR+) BC. We demonstrated that periodic fasting and MF enhanced the activity of tamoxifen and fulvestrant in HR+ mouse models by lowering circulating factors, such as IGF1, insulin, and leptin. Similarly, in HR+/HER2− BC patients receiving ET, MF cycles reduced blood glucose, serum IGF1, leptin, and C-peptide (a proxy of insulin production) and increased circulating ketone bodies (KBs), with leptin and IGF1 levels remaining low for extended periods. We found that combining ET and fasting upregulated the tumor suppressor PTEN and inhibited AKT and mTORC1 activity in HR+ BC cells. The transcription factor EGR1 mediated PTEN upregulation, inhibiting AKT as a result. We found MF to markedly delay acquired resistance to tamoxifen and fulvestrant in MCF7 xenograft-bearing mice. In addition, periodic MF cycles were as effective as the CDK4/6 inhibitor, palbociclib, in delaying fulvestrant resistance. The combination of fulvestrant, MF and palbociclib achieved powerful and long-lasting regression of MCF7 BC cell xenografts, and MF cycles could also reverse acquired resistance to the fulvestrant plus palbociclib regimen [58]. Our in vivo data in the mouse also suggested a striking ability of fasting/MF to avoid tamoxifen-induced endometrial hyperplasia, a relatively common side effect of this drug that can lead to bleeding and, in rare cases, to uterus cancer [58].

Previous studies had highlighted the ability of fasting or MF to strongly reduce the toxicity of several chemotherapeutics by inducing a state of increased resistance/fitness in healthy bodily tissues (as opposite to cancer cells, which were suggested to become more sensitive to chemotherapeutics) [59]. Thus, the available preclinical data strongly support clinical studies of fasting/MF in BC patients.

### Recent/Ongoing Clinical Studies of Fasting/MF in BC Patients

The safety, feasibility, and potential benefits of MF in patients with BC have been addressed in several clinical trials. In the DIGEST trial, which was a multicenter, randomized study, 131 patients with HER2-negative stage II/III BC, without diabetes and with a body mass index (BMI) over 18 kg/m^2^, were randomized to receive either a MF regimen or their regular diet for 3 days prior to and during neoadjuvant chemotherapy. This study failed to detect an effect of MF on the incidence of severe side adverse events or on the rate of pathological complete responses [60]. However, it did report a beneficial effect of MF on secondary endpoints, such as radiological as well as pathological tumor responses graded according to Miller and Payne and several patient-reported outcomes [60]. In addition, it also found MF to significantly reduce chemotherapy-induced DNA damage in patients’ lymphocytes [61]. Two other small randomized clinical trials, which included patients with gynecological cancers (including BC), also reported beneficial effects of a MF regimen on patients’ QoL [62,63].

A clinical trial conducted by our group at the IRCCS Ospedale Policlinico San Martino in Genoa [NCT03595540] and a similar study performed at the Fondazione Istituto Nazionale dei Tumori in Milan [NCT03340935] demonstrated tolerability and beneficial metabolic effects of MF in women undergoing ET for BC. These studies observed that MF was well-tolerated and recreated the metabolic changes seen in animal models. Specifically, MF led to significant reductions in insulin, IGF1, and leptin levels, with low levels of IGF1 and leptin persisting for extended periods, akin to findings in mice.

In the study conducted at the IRCCS Ospedale Policlinico San Martino, the incorporation of daily, “light” muscle training to enhance muscle anabolism, along with dietary recommendations (as per international guidelines for nutrition in cancer patients; [64]) to prevent malnutrition between MF cycles, likely contributed to beneficial changes in body composition observed in patients. These changes included an increase in phase angle and lean body mass, a reduction in fat mass, and stable body weight. Meanwhile, in a sub-analysis of the NCT03340935 clinical trial, fasting combined with chemotherapy improved survival in advanced triple-negative BC patients [65].

The NCT03454282 clinical trial, also known as DigesT Trial (Impact of Dietary Intervention on Tumor Immunity) showed several immunomodulatory effects induced by MF, including reductions in polymorphonuclear myeloid-derived suppressor cells and increases in CD8+ T cells, activated dendritic cells, natural killer (NK) cells, IFNγ, and M1-like macrophages.

Other ongoing clinical trials, such as the NCT03162289, NCT05259410, NCT04708860, NCT06123988, and NCT06106477, are still recruiting participants and aim to investigate the potential role of fasting or MF in BC patients.

Overall, the available results are particularly noteworthy given the concerns about using fasting/MF in oncology. They highlight the potential of MF to improve metabolic and immunological parameters in BC patients. However, it is crucial for patients to consult with their healthcare providers before making any changes to their dietary regimen. Large, randomized clinical trials are now needed to determine the effect of MF as an aid to standard therapies in BC patients.

## 4. The Role of CR in BC Treatment

CR involves reducing caloric intake without causing malnutrition and has been widely studied for its potential health benefits, including lifespan extension and disease prevention. Typically, CR reduces caloric intake by 20–40% from the standard diet, providing all essential nutrients while limiting calories. The concept of CR dates back to 1942, when Tannenbaum first demonstrated that CR could significantly reduce the development of mammary tumors in rodents [66], marking the beginning of extensive research into its cancer-protective effects. Both animal and human studies show that CR leads to significant, sustained metabolic, immunologic, and hormonal changes associated with a reduced BC risk as reviewed elsewhere [55,67,68,69]. These changes include lower insulin levels, improved insulin sensitivity, increased IGFBP-1 and sex hormone binding globulin (SHBG), decreased bioavailable testosterone and estrogen, reduced inflammation and oxidative stress, and enhanced anticancer immunity [55,67,68,69]. At the molecular level, long-term CR activates pathways associated with DNA repair, autophagy, and antioxidant responses while inhibiting several oncogenic pathways involved in cell proliferation and senescence [55,67,68,69,70].

### Clinical Insights and Challenges of CR in BC Management

Although there is abundant literature detailing the mechanisms and effects of CR in BC, its clinical utility is significantly limited by challenges in maintaining long-term adherence, as well as concerns regarding potential malnutrition and weight loss risks [67]. Nevertheless, a prospective case–control study conducted across two breast units in Italy involved 39 patients undergoing neoadjuvant chemotherapy who participated in a CR regimen. This regimen entailed a 30% reduction in caloric intake, which increased to 50% on chemotherapy days. In this small-scale study, combining CR with neoadjuvant chemotherapy significantly enhanced therapeutic outcomes, as evidenced by substantial reductions in tumor size and favorable lymph node status compared to neoadjuvant chemotherapy alone [71].

In another multicenter matched case–control study conducted in Spain, the association between BC risk and women’s relative caloric intake (actual/predicted intake) was analyzed, considering participants’ physical activity and basal metabolic rate. The results suggest that restricted caloric intake provides a protective effect against BC, particularly among premenopausal women, whereas excessive caloric consumption increases the risk, especially as observed among postmenopausal women and those with low adherence to the MedDiet. A positive association between relative caloric intake and BC risk was noted across all pathological subtypes (each 20% increase in relative caloric intake was associated with a 13% higher risk of HR+ tumors and a 7% increase for triple negative tumors) [72].

The Look AHEAD trial [NCT00017953], a randomized study of 4859 overweight, diabetic patients without a baseline cancer diagnosis, involved an intensive lifestyle intervention with a calorie goal of 1200 to 1800 kcal daily. After one year, the intervention group had an average weight loss of 8.7 kg compared to 0.75 kg in the control group. Over a median follow-up of 11 years, the intensive lifestyle intervention group had a 16% lower incidence of obesity-related cancers, such as those of the esophagus, colon, rectum, kidney, pancreas, stomach, liver, gallbladder, thyroid, uterus, ovary, postmenopausal breast, and multiple myeloma. This reduction was attributed to weight loss. No difference in the incidence of cancers not associated with obesity was documented [73]. Table 2 provides a summary of clinical studies investigating the effects of CR in BC patients.

## 5. Therapeutic Role of KDs in BC Management

The KD, a high-fat, carbohydrate-restricted diet without limiting calories, was first described in 1921 by Dr. Wilder, who studied this dietary approach as a means to treat epilepsy [74]. Lately, KDs have gained attention in oncology due to their potential modulation of cancer cell metabolism [75], promotion of favorable metabolic parameters associated with cancer treatment outcomes [75], synergy with anticancer therapies, and as a promising strategy to reduce treatment-related adverse events [76]. However, there are still several concerns regarding the potential adverse effects of these diets in cancer patients, including fatigue, decreased appetite, nausea, constipation, micronutrient deficiencies, hyperlipidemia, and unintended weight loss [77]. There are four different KDs, i.e., classic KD, the medium chain triglyceride (MCT) diet, the modified Atkins diet, and the low glycemic index treatment, which differ in the ratio of fat to carbohydrate and protein grams combined and in the type of fat used [78]. Thus, a 4:1 ratio is stricter than a 3:1 ratio and is, in general, more effective but more difficult to adhere to.

KDs mimic the metabolic state induced by starvation, forcing the body to use ketone bodies (KBs) derived from fatty acid β-oxidation as its primary source of energy. This is especially relevant in the case of BC, given that several types of BC are characterized by the Warburg effect [79]. Several mechanisms of action of KDs have been proposed to contribute to slow tumor progression. KDs increase blood KBs and reduce circulating glucose as well as IGF1 and insulin, growth factors which promote cell proliferation and escape from apoptosis [79,80]. In addition, KDs induce the cellular antioxidant system by activating the nuclear factor erythroid-derived 2-related factor 2 (Nrf2) [81], a main inducer of detoxification genes. KDs are able to blunt glycolytic activity, thereby reducing the acidity of the tumor microenvironment and reducing the availability of lactate as a substrate for biomass synthesis, leading to reduced metastases [80,82]. KDs could also modulate immune function as they were found to enhance the innate and adaptive immune responses against tumor cells, including cytolysis mediated by tumor-reactive CD8+ T cells [83], and to influence the activity of mammalian target of rapamycin (mTOR), which regulates the immune function [84]. Moreover, KDs can restore the Th17/Treg balance in patients with childhood intractable epilepsy [85]. A recent meta-analysis suggests that KDs lower inflammation by downregulating the levels of TNF-α, especially in people aged ≤ 50 years, and of IL-6, especially in subjects with a BMI greater than 30 kg/m^2^ [86]. KDs are able to potentiate the antitumor effects of chemotherapy and radiotherapy, reducing the needed dosages, thereby improving the QoL of cancer patients [87,88]. Finally, studies show that KDs help avoid high blood glucose and insulin levels in response to treatment with PI3K, AKT or mTOR inhibitor (the so-called insulin-feedback) and cooperate with these agents in achieving optimal tumor growth control [89]. For more detailed information on the mechanisms of action of KDs in the context of cancer, we invite the reader to see some of the comprehensive reviews recently published [90,91,92].

### Clinical Evidence of the Beneficial Role of the KDs in BC Patients

Several clinical studies aimed at exploring the tolerability and the potential beneficial effects of KDs in BC patients have been already completed or are currently ongoing (Table 3). 

In a RCT, patients with locally advanced or metastatic BC (MBC) undergoing planned chemotherapy were randomly assigned to a group receiving a KD or to a control group with a standard diet (SD) for 3 months (IR.SBMU.NNFTRI.REC.1396.187). The study showed that chemotherapy combined with KD could improve biochemical parameters (i.e., decreased fasting glucose), body composition (i.e., reduced BMI, body weight and fat %), and overall survival with no substantial side effects in lipid profile and kidney or liver damage markers [93].

In a similar study, patients with locally advanced or MBC were randomly assigned to either a KD or a control group for 12 weeks [IRCT20171105037259N2]. Compliance among KD subjects ranged from 66.7 to 79.2 %. The KD group showed higher global QoL and physical activity scores compared to the control group at 6 weeks but not at 12 weeks, although a significant inverse association was observed between total carbohydrate intake and serum β-hydroxybutyrate (BHB) at 12 weeks. The KD did not affect thyroid hormones, electrolytes, albumin, lactate dehydrogenase (LDH), or ammonia. The reduction in lactate and alkaline phosphatase (ALP) in the KD group suggests that a KD may benefit patients with BC. The KD group displayed a significant reduction in calorie intake, likely due to decreased appetite associated with ketosis [94]. Interestingly, it has been suggested that ketosis (i.e., BHB levels ≥ 0.5 mmol/L) [95] may enhance the effectiveness of chemotherapy while reducing the side effects of the treatment [96].

In a prospective, non-randomized, controlled phase I clinical trial [KETOCOMP study, NCT02516501], patients with non-MBC were allocated to either a KD or a SD during radiotherapy. The KD group displayed significantly higher fasting BHB than the SD group and a significantly decrease in free T3 levels. Remarkably, insulin and IGF1 decreased in both groups, but slightly more in the KD group. After an initial water loss, the KD tended to decrease body weight and fat mass while preserving fat-free and skeletal muscle mass. Importantly, no grade > 1 diet-related adverse events were reported by the patients. Global QoL remained stable in the SD group but increased in the KD group [97]. Furthermore, women on the KD experienced significant improvements in emotional and social functioning, sleep quality, their future perspectives, and systemic therapy side effects. Although breast symptoms rose significantly in both groups, the increase was less pronounced in the KD group. There were no signs of detrimental effects of the KDs on either liver or kidney function. In contrast, several biomarkers of non-alcoholic fatty liver disease (NAFLD) or metabolic syndrome significantly improved in the KD but not the SD group [98].

Another randomized trial enrolling 80 patients with locally advanced BC or MBC (RCT20171105037259N2) reported that a 12-week eucaloric MCT KD exerted beneficial effects by lowering insulin and TNF-α and upregulating IL-10. The KD also improved clinical response, as shown by a reduction in tumor size and a downstage in patients with locally advanced disease; however, no significant differences in response rate were observed in patients with MBC [99].

The KOLIBRI non-randomized trial [NCT02092753] showed that a healthy SD, a low-carb diet, and a KD for BC patients during the rehabilitation phase were feasible in daily life and resulted in enhanced QoL, body composition, and physical performance. KD participants displayed a very good physical performance and muscle/fat ratio. Despite increased cholesterol levels, KD patients had the best triglyceride/HDL ratio and homeostatic model assessment of insulin resistance index [100].

A recent non-randomized clinical trial [NCT03535701] assessed the feasibility and the sustained metabolic effects of a personalized well-formulated ketogenic diet (WFKD) consumed for 6 months in women diagnosed with stage IV MBC undergoing chemotherapy. The adherence to the WFKD was high and the diet was well tolerated, without no adverse diet-related events. Body weight decreased 10% after 3 months, mostly from body fat. In addition, fasting plasma glucose and insulin, as well as insulin resistance, diminished significantly after 3 months, an effect that persisted at 6 months [101].

The ongoing phase 2 clinical trial [NCT05090358] includes 15 patients diagnosed with stage IV BC, specifically metastatic BC with PIK3CA mutations. This randomized study aims to evaluate the effectiveness of a very-low-carbohydrate KD, a low-carbohydrate diet, and the study drug canagliflozin in preventing high blood sugar levels. The trial seeks to determine whether these interventions can enhance the efficacy of standard cancer treatments, alpelisib and fulvestrant. This trial is particularly noteworthy, supported by findings from Hopkins and colleagues, which suggest that the KD can be a powerful adjunct to PI3K inhibitors in PTEN/PIK3CA mutant endometrial patient-derived xenograft tumors, enhancing their therapeutic outcomes by preventing insulin feedback [89].

Although most of the available pre-clinical and clinical studies suggest that KDs are safe, tolerable, have beneficial metabolic effects, and, to some extent, potentiate the antitumoral effects of conventional anticancer therapies, there is still a great need for further larger randomized, controlled clinical trials to generate evidence for the beneficial effects on biochemical parameters and body composition and the antitumor effects of KDs, administered alone or together with standard or targeted treatments, to allow recommendations for dietary modifications for BC cancer patients aimed at improving survival, QoL, and prognosis. This strong evidence is also needed to eliminate the concerns regarding the safety and long-term effects of sustained ketosis in subjects diagnosed with BC. Until then, caution should be implemented in the use of KDs, and physicians should keep prioritizing evidence-based, standard-of-care therapies.

## 6. Vegan or Plant-Based Diets: Promising Approaches to BC Management

A vegan diet strictly excludes all animal products, including meat, dairy, eggs, and honey, emphasizing instead the consumption of plant-based foods such as fruits, vegetables, legumes, nuts, seeds, and whole grains. This dietary choice is motivated by ethical considerations, environmental sustainability, and potential health benefits. The modern vegan movement was formally established in 1944 by Donald Watson and a group of non-dairy vegetarians in the UK, who coined the term “vegan” and founded the Vegan Society [102]. In contrast, a plant-based diet primarily focuses on foods derived from plants while limiting or avoiding animal products and processed foods [103]. Recent studies suggest that both vegan and plant-based diets may offer significant benefits in BC prevention and management [103,104,105]. A key mechanism is the reduction in exposure to carcinogens, as vegan diets eliminate processed meats and other animal products that can contain harmful compounds such as heterocyclic amines and polycyclic aromatic hydrocarbons, which form during high-temperature cooking [106,107]. Plant-based foods are rich in antioxidants and phytochemicals like flavonoids and carotenoids, which protect cells from DNA damage and reduce inflammation, potentially lowering cancer risk [17]. A fiber-rich vegan diet or plant-based diet also promotes a healthy gut microbiome, which plays a role in modulating the immune system and reducing inflammation, both critical factors in cancer prevention and management [45]. Maintaining a healthy weight is crucial in reducing the risk of BC recurrence, and vegan or plant-based diets, typically lower in calories and higher in fiber, can aid in weight management [108].

### Clinical Studies Involving Vegan or Plant-Based Diets in BC Patients

Previous research on vegan diet or plant-based diets and BC patients has been inconsistent, with some studies finding reduced risk and others showing no significant associations [109,110]. Table 4 summarize clinical studies investigating the effects of vegan diet or plant-based diets in BC patients. 

Epidemiological studies have shown that populations adhering to plant-based diets have lower rates of BC compared to those consuming a typical Western diet. For instance, the Plant-based Diets and Risk of Cancer in the Adventist Health Study-2 (AHS-2) clinical trial [NCT03615599] analyzed data from approximately 96,000 participants, a large and diverse group that provided a robust dataset for examining the impact of diet on BC risk. The study classified subjects into five dietary patterns: vegan, lacto-ovo-vegetarian, pesco-vegetarian, semi-vegetarian, and non-vegetarian, using a validated food frequency questionnaire. Among the 50,404 female participants, including 26,193 vegetarians, 892 incident BC cases were identified, with 478 cases among vegetarians. The results showed that vegetarians as a whole did not have a significantly lower risk of BC compared to non-vegetarians. However, vegans exhibited consistently lower, though not statistically significant, risk estimates compared to non-vegetarians [111].

In two large prospective cohort studies in the US conducted by Romanos-Nanclares and colleagues, the impact of different types of plant-based diets on BC risk was examined. The study found that a healthful plant-based diet, rich in antioxidants, vitamins, and dietary fiber, may lower the risk of ER-negative BC. These types of cancer are less influenced by hormonal factors and may be more affected by diet. The potential mechanisms include antioxidant and anti-inflammatory effects, improved insulin resistance, and glycemic control [105]. These findings suggest that other components in plant foods, such as flavonoids and phenolic compounds, might contribute to the observed benefits. In contrast, less healthy plant-based foods, such as refined grains, pastries, sugary beverages, and processed foods, have been linked to higher BC risk in other studies [112,113,114].

A case study investigated how different plant-based diets affect BC risk. The study found that women who adhere to healthy plant-based diets, rich in fruits, vegetables, whole grains, and legumes, have a lower risk of developing BC compared to those who consume less nutritious plant-based foods. The findings highlight the potential of high-quality, healthful plant-based diets as a preventative measure against BC. This study emphasizes the critical role of diet quality in cancer prevention [115].

In a study involving 3,646 BC survivors, researchers found that adherence to healthful plant-based eating patterns after diagnosis reduced the risk of non-BC mortality. Conversely, greater adherence to unhealthful plant-based eating patterns increased the risk of non-BC mortality. No associations were found between plant-based eating patterns and BC recurrence or BC-specific mortality. This highlights the importance of the quality of plant foods in achieving a healthful dietary pattern, which can potentially improve overall survival in BC survivors [104].

The RCT [NCT03045289] demonstrated that a whole-food, plant-based diet could lead to beneficial outcomes in women with metastatic BC, particularly in terms of weight management (significant weight loss), cardiometabolic health (evidenced by reductions in total and LDL cholesterol, lower blood pressure, and improved glycemic control), and hormonal balance (reduced levels of IGF1 and alterations in estrogen levels) [116].

The Intravenous Chemotherapy and Plant-based Dietary Supplements (CIVCAP) trial [NCT03959618] was designed to determine the quantitative impact of the use of plant-based dietary supplements by BC patients treated in a neo-adjuvant and/or adjuvant condition by IV chemotherapy.

Furthermore, studies suggest that BC survivors who adopt a vegan or plant-based diet may experience improved survival rates. The Women’s Healthy Eating and Living (WHEL) study [NCT00003787], although not exclusively vegan, indicated that a diet high in vegetables, fruit, and fiber and low in fat could improve survival rates in BC survivors [117].

Bu et al.’s systematic review analyzed recent studies on the influence of dietary patterns on BC, aiming to identify diets that could prevent BC, improve prognosis, and enhance the QoL for BC survivors. They found that plant-based diets were linked to a reduced risk of invasive BC and improved overall survival, particularly in ER-negative, HER2 basal-like, and luminal A BC patients. The mechanisms include reductions in IGF1, blood glucose, and cholesterol and the beneficial effects of various phytochemicals [118].

Overall, for those considering adopting vegan or plant-based diets for BC prevention or as part of their treatment plan, it is essential to consult with healthcare providers, including oncologists and registered dietitians, to ensure that the dietary plan is nutritionally adequate and tailored to individual health needs. Emphasizing whole, minimally processed plant-based foods can maximize nutrient intake and health benefits. Diversifying the diet to include a variety of fruits, vegetables, legumes, nuts, seeds, and whole grains helps meet nutritional needs and avoid deficiencies. Necessary supplements, such as vitamin B12, which is not naturally present in plant-based foods, and possibly vitamin D and omega-3 fatty acids, should be considered based on dietary intake and their blood levels [119]. Being updated with the latest research on vegan or plant-based diets and cancer, while remaining receptive to adjusting dietary practices based on emerging scientific evidence, can further enhance the benefits of these diets in the context of BC [120].

Finally, it is crucial to emphasize that dietary regimens such as fasting/MF, CR, KD, and vegan or plant-based diets often lead to weight loss or reduced protein intake, which may be detrimental in cancer patients, particularly in those at risk for malnutrition [121,122]. Thus, we remind the reader that until new, strong evidence in support of the use of these types of diets becomes available, respecting international guidelines for nutrition in cancer patients remains essential [123] and that fasting/MF, CR, KD, and vegan or plant-based diets should only be recommended within clinical trials.

## 7. Antitumor Mechanisms and the Impact of Physical Activity on BC: Preclinical and Clinical Insights

The compelling data produced to date suggest that structured physical activity/exercise training decreases the risk of developing some types of cancer, helps cancer survivors cope with and recover from anticancer therapies, improves the long-term health of cancer survivors, and could even reduce the risk of recurrence and prolong survival in some cancer survivor groups [124,125]. In BC patients, physical activity, both before and after the cancer diagnosis, has been associated with a lower risk of disease recurrence and reduced overall and cancer-specific mortality compared to their sedentary counterparts [124]. Clinical studies conducted or currently ongoing in BC patients examining physical activity are summarized in Table 5. 

Additionally, in BC survivors, structured exercise improves lymph drainage from their upper limbs, thereby reducing mastectomy side effects, lowers their risk of cancer relapse, and enhances their immune functions [126]. Moreover, moderate-intensity aerobic exercise combined with resistance training benefits early adjuvant BC treatment, positively affecting physical fatigue, muscle strength, and cardiopulmonary function [127]. Structured exercise interventions also impact biomarkers of cancer risk [128]. Consistent with clinical epidemiological data, several preclinical studies have shown that exercise significantly inhibits tumor growth and metastatic dissemination in BC human and murine models [129,130,131]_ENREF_3. However, others have reported no significant differences [132,133,134,135]_ENREF_15 or even increased disease incidence and worse prognosis [136,137]. These discrepancies might be explained by differences in tumor model (breast vs. other cancer types), exercise paradigm (forced vs. voluntary exercise), stimulus (short-term vs. long-term training), site of tumor implantation (orthotopic vs. subcutaneous), and dietary background (high-fat vs. low-fat diet). Notably, prolonged, intense exercise results in immunosuppression [138], and the relationship between immunosuppression and cancer is well-established. Similarly, intense long exercise can induce higher levels of pro-inflammatory cytokines, potentially increasing the risk of chronic inflammation [138], which is known to promote cancer progression. On the other hand, this type of exercise can also lead to elevated glucocorticoid levels that may act as anti-inflammatory signals to cells like macrophages, inhibiting their migration towards sites of inflammation [138].

Various mechanisms have been proposed to underlie the protective effects of physical activity against cancer, including reduced body fat, enhanced gut motility, decreased lifetime exposure to estrogen and other hormones, improved antioxidant defenses, and promotion of anti-tumor immune defenses [14] (Figure 2). In cancer survivors, physical activity mitigates lymphedema [139] and aids recovery after chemotherapy [140]. Indeed, specific types of exercise might be recommended to cancer patients experiencing peripheral neuropathy, a common side effect induced by chemotherapeutic agents [140]. Furthermore, exercise training mitigates anemia, a hematological condition associated with poor prognosis in cancer patients [141], and enhances sensitivity to immunotherapy by improving body composition [142]. Physical activity exerts beneficial effects at both early and advanced cancer stages [143], as well as during the preoperative period. However, it is important to note that the mechanistic properties of exercise on tumor biology and progression are likely different in the pre- vs. post-diagnosis setting.

The primary hypothesis regarding the positive effects of long-term or regular exercise training on controlling cancer progression has centered on reducing the basal systemic levels of cancer risk factors [14,144]. However, it is also important to consider the significant acute increase in several potential anticancer systemic factors during each bout of exercise (see Dethlefsen et al. [145]). Mechanisms frequently studied in exercise oncology include sex hormones, insulin-related pathways, low-level chronic inflammation, immune function, adipokines, and myokines. Nevertheless, the relative influence of each mechanism and their combined effects on cancer survival still need to be fully elucidated. Table 6 summarizes the preclinical studies of exercise in human and murine breast cancer models.

### 7.1. Modulation of Circulating Hormones

#### 7.1.1. Sex Steroid Hormones

Elevated estrogen, progesterone, and androgen levels, along with lower SHBG, are implicated in BC development [175]. Sex steroid hormones are linked to the association between physical activity and BC risk [176]. Lack of structured exercise boosts aromatase activity, primarily in postmenopausal women, increasing circulating estradiol, a key BC risk factor [176]. Excessive estrogen production by adipose tissue is a principal mechanism linking obesity to BC [177]. Recent findings indicate that women who engage in structured exercise (corresponding to 3 h per week of brisk walking) after BC diagnosis have a significantly lower risk of death or BC recurrence than women who were physically inactive [14,178]. Overweight or obese sedentary BC survivors benefit from combined aerobic and resistance exercise, which decreases estradiol and increases SHBG levels, leading to changes in body composition, such as reduction in fat mass and an increase in lean mass [179]. Systemic sex hormone levels correlate tightly with body composition in postmenopausal women (e.g., BMI) [180]. Exercise-induced reductions in sex hormone levels are mainly observed in overweight women who lose weight during intervention [181]. However, a large randomized trial on premenopausal women failed to show changes in sex hormone levels, likely due to the lack of weight loss [182], though cross-sectional studies suggest an inverse correlation between physical activity and estradiol and testosterone levels [183]. Recent meta-analyses confirm exercise-induced decreases in sex hormone levels, regardless of menopausal status, supporting the role of physical activity in preventing BC [184].

A Mendelian randomization study using data from the UK Biobank demonstrated that physical activity is inversely associated with the risk of breast and colon cancer, independent of its effect on adiposity. The study also showed that the association between physical activity and cancer incidence at ten different sites is independent of BMI [185].

Swain et al.’s systematic review and meta-analysis support the role of physical activity in reducing circulating sex steroid hormones and increasing SHBG, suggesting its causal role in preventing BC [186]. They found no significant differences in hormone responses to physical activity between pre- and postmenopausal women, with both aerobic and resistance exercises showing effects on sex hormones [186]. Another recent systematic review and meta-analysis noted that, in premenopausal women, there is little evidence linking levels of estrogen, progesterone, or SHBG with BC risk, while androgens are positively associated. However, in postmenopausal women, higher estrogen and androgen levels increase BC risk, whereas higher SHBG levels decrease it, indicating notable dose–response relationships between sex steroid hormone concentrations and BC risk. These findings imply a role for sex steroid hormones in mediating the link between physical activity and BC risk [187].

Physical activity is proposed to decrease sex steroid hormones through various mechanisms. Low physical activity levels are associated with increased adiposity and enhanced aromatization of androgens to estrogens [188]. In line with these findings, it has been suggested that the effect of exercise on sex hormones is at least partially achieved by fat loss [189]. Physical inactivity also elevates inflammation, promoting aromatase upregulation and estrogen production [190,191]. Conversely, increasing physical activity stimulates the release of anti-inflammatory cytokines and reduces pro-inflammatory cytokine production. Exercise-induced upregulation of SHBG [186] may decrease sex hormone bioavailability [192], partly by reducing insulin levels and enhancing hepatic SHBG synthesis [193]. Furthermore, physical activity enhances insulin sensitivity and lowers insulin resistance (see below), effects observed with both regular activity and acute exercise bouts [194].

#### 7.1.2. Insulin/Insulin like Growth Factor 1

Insulin resistance has been associated with an increased risk of BC, higher rates of cancer recurrence, and poorer survival outcomes [195,196]. Insulin can directly promote tumor development by stimulating cell proliferation and activating the IGF1 system, which controls cell differentiation, proliferation, and apoptosis [197,198]. Additionally, insulin can modulate the synthesis, availability, and effects of sex hormones [199]. Regular physical activity improves insulin sensitivity and mitochondrial function, increases mitochondrial biogenesis, and reduces circulating glucose, insulin, and IGF1 levels by enhancing muscle glucose uptake [200]. This is likely due to the increased expression and function of the GLUT4 glucose receptor and various metabolic genes in physically active skeletal muscle [201,202]. Physical activity can also increase insulin-like growth factor-binding protein 3 (IGFBP-3) [203], which binds to IGF1, reducing its bioavailability. In a rat model of BC, exercise reduced circulating insulin and IGF1 levels and elevated plasma corticosterone, suggesting that exercise may exert anticancer effects by controlling glucose homeostasis [204]. However, clinical data on the effects of physical activity on the insulin pathway are inconsistent. In overweight or obese BC survivors, a 16-week program of combined aerobic and resistance exercise, performed three times per week, resulted in decreased circulating insulin and IGF1 levels, increased IGFBP-3 levels, improved metabolic syndrome markers, and a 4 kg reduction in body weight [179]. A meta-analysis by Kang et al. indicated that the reduction in fasting insulin levels achieved through exercise interventions in BC patients depended on weight loss [205]. Conversely, other studies have reported no changes in fasting insulin levels with exercise training, even when weight loss occurred [206,207]. Another large meta-analysis found no effect of exercise training on fasting insulin levels in healthy individuals without comorbidities (e.g., type 2 diabetes, metabolic syndrome) [208]. A recent meta-analysis suggested that physical activity interventions reduced fasting insulin, insulin resistance, and fasting glucose levels and elevated IGF1, but had no clear effect on IGFBP-3 or the IGF1:IGFBP-3 ratio. Strong evidence was established only for reductions in fasting insulin and insulin resistance [193]. Similarly, the results of studies on the effects of exercise on the IGF1 axis in cancer survivors are inconsistent [209,210]. However, a recent systematic review of 15 randomized controlled trials found that exercise training significantly improved metabolic biomarkers, including insulin, glycemic, and lipid profiles [211].

### 7.2. Inflammation and Immunity

#### 7.2.1. Cytokines and Adipokines

It is widely recognized that exercise modulates inflammatory responses [212,213,214], with lower levels of physical activity being associated with an adverse, chronic inflammatory profile [212,215]. Conversely, regular physical activity induces expression of anti-inflammatory cytokines and suppresses the expression of pro-inflammatory cytokines [214,216,217]. Inflammation stimulates cell proliferation and induces oxidative stress and detrimental changes in tumor microenvironment, leading to tumor initiation, promotion, malignant conversion, invasion, progression, and metastasis [218,219]. Hence, chronic inflammation is widely recognized as an underlying mechanism for carcinogenesis [14,190,220,221,222]. The effects of physical activity on systemic levels of pro-inflammatory cytokines are inconsistent, likely due to differences in adherence, population characteristics (active vs. inactive individuals), type of exercise (e.g., aerobic vs. resistance or a combination), and duration of the intervention, among other factors. Some studies have reported that physical activity decreases systemic CRP levels, but the degree of reduction depends on the intensity and duration of exercise, as well as the presence of obesity or low-grade inflammation (see Dethlefsen et al. [145]). A recent systematic review shows that cancer patients engaged in various exercise regimens exhibited improvements in immunity, including reductions in TNF-α, CRP, IL-8, and IL-6, along with an increase in NK cells [223]. In line with these results, regular daily physical activity has been recently associated with lower CRP levels [224]. In contrast, a meta-analysis of 160 exercise intervention studies showed no effect on the levels of CRP, IL-6, or TNF-α [208]. In most studies involving BC patients, there were no observed changes in CRP during the training period [225,226]. Similarly, a meta-analysis of the effects of exercise on BC survivors reported no alterations in CRP, IL-6, or TNF-α, nor in IL-8, Il-2, IL-10, or adiponectin [227]. However, a recent randomized trial documented a reduction in circulating levels of hs-CRP and IL-6 [228]. A prospective study noted lower CRP levels in BC patients engaged in moderate–vigorous physical activity compared to those with less activity [229]. In overweight or obese sedentary BC survivors, participating in combined aerobic and resistance exercise three times per week for 16 weeks resulted in decreased circulating hs-CRP, IL-6, IL-8, and TNF-α. These changes were observed at the end of the intervention and persisted three months after completing the exercise program, accompanied by a 4 kg weight loss [179]. Similarly, obese postmenopausal BC survivors undergoing supervised combined aerobic and resistance exercise sessions three times per week for 16 weeks exhibited higher plasma adiponectin and lower plasma CRP, leptin, IL-6, and IL-8 compared to baseline and a control group [230]. However, another meta-analysis of studies on BC survivors found no effect of exercise training on systemic TNF-α and IL-6 [205].

Resistance exercise may also benefit BC patients by reducing sarcopenic obesity. A high prevalence of sarcopenic obesity has been reported in BC survivors [179]. In postmenopausal women, sarcopenic obesity is associated with elevated pro-inflammatory mediators such as CRP, TNF-α, and IL-6 [231]. Conversely, sarcopenic obesity is an independent predictor of cancer survival [232]. In overweight or obese sedentary BC survivors, participating in combined aerobic and resistance exercise three times per week for 16 weeks led to significant reductions in biomarkers of sarcopenic obesity, including appendicular skeletal muscle index and BMI, compared with baseline and a control group [179]. Regular exercise induces significant reductions in abdominal and visceral fat, even without any accompanying body weight loss, irrespective of sex and age [233]. This decrease in fat mass leads to an increase in circulating adiponectin levels while simultaneously reducing several circulating pro-inflammatory adipokines, including IL-6, TNF-α, RBP-4, and leptin [234], many of which are implicated in the development and progression of BC [235]. Obesity-associated hyperleptinemia has been suggested as an important mediator in the pathophysiology of BC [108]. Thus, leptin, like insulin, acts as a growth factor for BC cells and attenuates apoptosis of BC cells [236]. In overweight or obese sedentary BC survivors who had recently completed treatment, participating in combined aerobic and resistance exercise three times per week for 16 weeks led to a down-regulation of serum leptin and an up-regulation of adiponectin, along with a decrease in the percentage of body fat. These changes were observed at the completion of the intervention and at follow-up, three months later [179]. A recent meta-analysis of randomized controlled trials assessing the effects of diet and exercise-induced weight loss on biomarkers of inflammation in BC survivors found that leptin levels were significantly reduced in the exercise-only group compared with the sedentary control, suggesting that leptin may be a primary mediator of exercise-induced improvements in BC recurrence [227]. In a randomized clinical trial conducted in premenopausal women at risk for BC, a dose–response effect of exercise on adiponectin, which increases, and leptin, which decreases, was observed, with this dose–response being dependent on changes in body fat [237]. Adiponectin exerts anti-tumor effects by suppressing cell proliferation, inhibiting tumor growth, increasing apoptosis, and inhibiting angiogenesis through multiple pathways [238]. Low levels of adiponectin, characteristic of obesity, are linked to increased proliferative activity, resulting in an enhanced risk of developing BC. Additionally, low circulating adiponectin is associated with a larger tumor size and poorer prognosis of BC [239]. Studies have shown that combined resistance and aerobic exercise, undertaken three times per week for 16 weeks, led to a marked increase in circulating adiponectin in overweight or obese BC survivors, accompanied by a decrease in fat mass and an increase in lean mass, with these changes persisting at the three-month follow-up [179]. Comparable results were obtained in a similar study involving obese postmenopausal BC survivors [230]. In line with these findings, a RCT conducted in overweight/obese BC survivors, involving 12 weeks of high-intensity interval training or moderate-intensity continuous training, increased adiponectin levels and reduced leptin levels. These changes were accompanied by decreases in body mass, fat mass, TNF-α, and IL-6 and an elevation of IL-10 [240]. Conversely, in a trial conducted in survivors of triple-negative BC with BMI > 25, moderate-intensity aerobic exercise (150 min per week for 12 weeks) did not change serum adiponectin, leptin, insulin, CRP, IL-6, or TNF-α, despite a loss of body fat, although serum leptin and adiponectin and their ratio significantly correlated with BMI [241]. However, a recent systematic review has shown that 12 controlled randomized clinical trials reported that exercise interventions significantly improved inflammation and immune response biomarkers, including leptin, adiponectin, TNF-α, IL-6, IL-10, and CRP [211].

#### 7.2.2. Immunity

In addition to inflammation, immune function is also modulated by physical activity [242,243]. Higher levels of physical activity have been associated, in a dose–response manner, with lower counts of white blood cells, basophils, monocytes, neutrophils, eosinophils, and lymphocytes, as well as with reduced levels of fibrinogen and a lower prevalence of clinically elevated CRP (although not immunoglobulin E, IgE). Interestingly, a positive association between circulating fibrinogen and BC risk has been reported [244]. Exercise training also leads to a down-regulation of the expression of Toll-like receptors (TLRs) on monocytes and macrophages [214], resulting in the mitigation of downstream responses, including the production of pro-inflammatory cytokines and the expression of the major histocompatibility complex and co-stimulatory molecules [214]. Preclinical studies have shown the inhibition of monocyte and macrophage infiltration into adipose tissue and the phenotypic switching of macrophages within adipose tissue [245]. Furthermore, human studies have found a decline in the circulating numbers of pro-inflammatory monocytes (CD14+CD16+) and an increase in the circulating numbers of regulatory T cells (TReg cells) [246]. Moreover, exercise training (combined aerobic and resistance exercise) has been shown to reduce the production of intracellular pro-inflammatory cytokines by monocytes, especially IL-1β, in women with stage I, II, or III BC within one year of completing treatment [247]. However, this study did not find changes in the expression of TLR2 and TLR4 on monocytes over the course of training, although it did report a downregulation of these receptors following acute exercise [247]. A higher percentage of CD14++CD16+ monocytes has been reported in BC patients, especially in those with stage III and IV disease, regardless of ER status [248,249].

Given that macrophages are key contributors to adipose tissue inflammation in obesity, it has been postulated that adipose tissue macrophages (ATMs) may link obesity to poorer cancer outcomes. Obesity promotes a phenotypic change in ATMs’ polarization, from a predominance of M2, “alternatively activated” ATMs, which act primarily in an anti-inflammatory fashion, to an increase in the prevalence of M1, “classically activated” ATMs, which are considered pro-inflammatory [250]. M1 ATMs release TNF-α and IL-6, which not only contribute to insulin resistance but are also associated with cancer recurrence [251,252]. M1 ATMs also produce paracrine and angiogenic factors that can support survival of damaged cells and actively stimulate tumor growth [253]. Exercise suppresses macrophage infiltration into adipose tissue, accelerates the phenotypic switching from the pro-inflammatory M1-type to the anti-inflammatory M2-type macrophages, and suppress M1-type macrophage infiltration into adipose tissue, as suggested by a preclinical study [245]. In obese postmenopausal BC survivors, 16 weeks of supervised combined aerobic and resistance exercise sessions three times per week elicited a decrease in the number of M1 macrophages and an increase in the number of M2 macrophages in adipose tissue, accompanied by higher plasma adiponectin levels and lower plasma CRP, leptin, IL-6, and IL-8 compared to baseline as well as to the control group [230].

Exercise can enhance anticancer immunity and mitigate the pro-tumoral effects of immunological senescence [254]. Among immune cells, NK cells are the most sensitive to exercise and can be promptly mobilized during physical activity [255]. Exercise mediates NK cell mobilization and trafficking through IL-6-dependent and adrenaline-dependent pathways, thereby limiting tumor growth [256]. Indeed, blocking β-adrenergic signaling diminishes the tumor-suppressive effect of exercise. Preclinical studies suggest that NK cell function facilitates the protective effects of exercise BC prevention. Thus, exercise training has been shown to increase NK cell trafficking and infiltration into tumors in vivo [256]. It has been suggested that higher-intensity exercise acutely induces greater mobilization and larger changes in NK cell cytotoxicity compared to lower-intensity exercise [257]. However, a recent study suggests that exercise intensity may not significantly impact changes in resting NK cell function among women at high risk for BC [258]. The cytotoxic activity of NK cells against various cancer cell lines increases in response to both acute and chronic exercise, attributed to phenotypic shifts in NK cells and their frequent mobilization and redistribution with each bout of exercise [257]. A recent case–cohort study reported an inverse correlation between BC risk and higher resting levels of NK cells among postmenopausal women, although this association did not reach statistical significance [259].

Similarly, T cells, the primary cells for adaptive immunity with direct antitumor actions, are also regulated by exercise [256]. A study conducted by the research team at the University of Turku in Finland revealed that just 10 min of moderate-intensity exercise can increase the total leukocyte count in BC patients by 29%. This includes CD8+ T cells, CD19+ B cells, CD56+ CD16+ NK cells, and CD14+ and CD16+ monocytes. Notably, CD8+ T and NK cells are crucial for killing tumor cells, while other immune cells, such as CD4+ T cells, enhance their cytotoxicity [260].

Exercise-dependent CD8+ T cell cytotoxicity mediates the attenuation of tumor growth [148]. Consistent with these findings, in a murine model of BC, exercise training increased the number and effector function of CD8+ T cells in tumor tissue through CXCR3 signaling, enhancing the anticancer activity of immune checkpoint blockade (i.e., anti-PD-1 alone or in combination with anti-CTLA-4) [160].

### 7.3. Epigenetic Effects of Exercise on Cancer

Several studies have suggested that physical exercise and diet can modulate both DNA methylation status and miRNA expression, potentially impacting BC risk [261,262]. Similarly, various studies have demonstrated the protective role of exercise in reducing BC incidence and mortality by inducing genetic and epigenetic modulation [263,264,265,266].

#### 7.3.1. Genomic DNA Methylation Status

One of the hallmarks of cancer is an altered overall genomic methylation status, particularly characterized by hypomethylation in repetitive elements [267]. It has been suggested that physical activity may increase the levels of global genomic DNA methylation, partially restoring the hypomethylated genome observed in cancer [268]. Additionally, hypermethylation, especially in cytosine within CpG dinucleotides in gene promoters, has been associated with neoplastic mutations [269]. Physical activity has been linked to decreased estrogen levels, which can induce promoter hypermethylation of tumor suppressor genes implicated in BC tumorigenesis [270]. Furthermore, physical exercise promotes the expression of the tumor suppressor protein p53, which is often down-regulated in various cancers, through epigenetic mechanisms involving miRNAs [271].

#### 7.3.2. MicroRNAs (miRNAs)

_ENREF_67Physical activity influences the expression of miRNAs, impacting various processes such as inflammation, stress response, treatment sensitivity, and treatment-related side effects [272]._ENREF_71 Consequently, a growing body of evidence suggests the involvement of miRNAs in exercise adaptation and the protective effects of exercise training across various pathologies, including cancer (refer to Silva et al. [273]; Masi et al. [274]; Domańska-Senderowska et al. [275]; and Orlandella et al. [276])._ENREF_74_ENREF_75_ENREF_72_ENREF_76 Notably, circulating miRNAs respond to different types and durations of exercise, including acute, chronic, aerobic, vigorous, and resistance training [277]. Importantly, miRNAs are dysregulated in all types of cancers [278,279,280]_ENREF_80, playing roles in crucial signaling pathways for cancer development (e.g., sustaining cell cycle progression, proliferation, survival, invasion, metabolism, angiogenesis, and metastasis) and in resistance to various therapies [281] (refer also to He et al. [282]_ENREF_87), exhibiting either pro- or anti-tumor effects [283]. They can be categorized as oncogenes (oncomirs), tumor-suppressor genes, pro-metastatic (‘metastamiRs’), and metastasis-suppressor miRNAs (see van Schooneveld et al. [280]). Therefore, it has been suggested that identifying miRNAs affected by physical activity/exercise training could be beneficial in oncology, not only to enhance physical performance and recovery during and after cancer treatments but also to regulate aberrant signaling pathways implicated in tumorigenesis [284,285]._ENREF_94

In the case of BC, a recent study analyzing eight GEO DataSets miRNA expression datasets and the Cancer Genome Atlas Breast Cancer (TCGA BRCA) dataset revealed that 30 miRNAs were significantly upregulated in BC samples compared to the controls, while 19 were downregulated [266]. Consistent with these findings, the study by Falzone et al. reported that several of the miRNAs modulated by exercise training, such as the upregulated hsa-miR-140-5p, hsa-miR-7-5p, and has-miR-590-5p and the downregulated hsa-miR-139-5p, hsa-miR-199a-5p, and hsa-miR-92a-3p, exhibit clear interactions with numerous epithelial-to-mesenchymal transition (EMT) genes that display aberrant expression in BC samples. These genes are also associated with changes in miRNA expression in BC tissues compared to healthy tissue, including ZEB1, ZEB2, VIM, and SNAIL2 [266]. This implies that precise regulation of these miRNAs achieved through exercise interventions may inhibit EMT, thereby reducing the risk of BC recurrence [286].

Regular exercise training has been shown to reduce the circulatory level of miR-21 [287,288], a well-known tumor promoter whose serum levels are elevated in the plasma and tumor tissue of BC patients [289]. MiR-21 has been suggested as a negative prognostic factor for BC, as its higher expression in invasive ductal carcinoma compared to normal breast tissue is positively associated with tumor size, stage, grade, and Ki-67 expression, as well as with ER negativity and HER2 positivity and lower overall survival rates [290]. Similarly, miR-21 is inversely correlated with disease-free survival in heterogeneous BC [291]. Conversely, interval exercise (treadmill running at gradually increasing speeds, 5 days/week for 5 weeks) induces an upregulation of miR-206 and let-7 (two tumor suppressors) and a downregulation of miR-21 expression in a murine model of BC. These changes are accompanied by a reduction in tumor size and the expression of ER-α, HIF-α, VEGF, TNF-α, CD31 (a marker of angiogenesis), and Ki67 (a marker of proliferation) in tumor tissue, as well as in the serum levels of estradiol (E2), and an increase in the mRNA levels of PDCD-4 (a tumor suppressor and target of miR-21). Consistent with these findings, Khori et al. demonstrated synergistic effects of exercise and tamoxifen in decreasing tumor growth via inhibition of miR-21 expression [287]. They also reported a reduction in serum estradiol and ER-α expression and in tumor levels of IL-6, NF-kB, and STAT3, as well as an upregulation of TPM1 and PDCD4 expression [287]. Overall, these preclinical studies suggest that the antitumoral effects of exercise training could be mediated in part by the modulation of several miRNAs, which in turn leads to decreased angiogenesis and increased apoptosis.

A randomized trial conducted in healthy women and in BC patients undergoing hormonal therapy (letrazole or tamoxifen) studied the effects of high-intensity aerobic interval training (38 min, three times/week for 12 weeks) on the expression of circulating miRNAs. BC patients displayed higher levels of five oncomiRs (miR-21, miR-155, miR-221, miR-27a, and miR-10b) and lower levels of five tumor suppressor miRNAs (miR-206, miR-145, miR-143, miR-9, and let-7a) compared to healthy controls. Hormonal therapy reduced the expression of oncomiRs and increased tumor suppressor miRNAs. In addition, the combination of exercise training and hormonal therapy elicited a stronger downregulation of these oncomiR and a stronger elevation of tumor suppressor miRNAs compared to BC patients in hormonal therapy alone [292].

Another trial conducted with post-menopausal BC survivors reported a decrease in serum levels of miR-106b (a potential prognostic marker of BC recurrence), miR-27a, and miR-92a, accompanied by an increase in miR-191, let-7b, and miR-24 after 6 months of exercise and weight loss [264]. It is worth noting that serum levels of miR-92a are elevated in BC patients [293], and miR-106b is also upregulated in BC [266]. The study by Adams et al. also documented a negative correlation between body mass index (BMI) and serum levels of miR-191, miR-17, miR-103a, and miR-93, and a positive association between BMI and miR-22, miR-122, miR-126, and miR-150 [264]. Ingenuity pathway analysis identified “estrogen-mediated S-phase entry” and “molecular mechanisms of cancer” as the top canonical pathways significantly correlated with BMI-associated and intervention-responsive miRNAs. These pathways contain obesity- and cancer-relevant genes, including the E2F family of transcription factors and CCND1, which have been implicated in sporadic BC [264]. Olson et al. conducted a clinical trial in metastatic BC patients and found that after 12 weeks of nutrition and exercise interventions (150 min of moderate physical activity and strength training two times/week), three known tumor suppressor miRNAs (miR-10a, miR-211, and miR-205) were significantly upregulated in the serum of high responders post-exercise intervention compared to their baseline and compared to non-responders [294].

## 8. Evidence on the Impact of Integrated Dietary and Physical Activity Interventions in BC Treatment

Cancer prevention strategies, including dietary interventions and physical activity, should be integrated into treatment plans for patients with BC and the general population, as their regular application can reduce the individual risk of developing certain chronic diseases. Several clinical trials have evaluated the use of different dietary regimens and physical activity as a combined lifestyle intervention against BC.

An instance from a non-oncological context is the 3-month randomized controlled trial [NCT04004403] investigating the comparative effects of fasting coupled with exercise among patients with non-alcoholic fatty liver disease [295]. Herein, eighty patients were allocated randomly to each singular intervention, their combination, or a control cohort. Results revealed that the combined interventions yielded a reduction in intrahepatic triglyceride content compared to exercise alone and control groups, though not when compared to fasting alone. These interventions elicited significant reductions in body weight, fat mass, waist circumference, ALT levels, insulin, and insulin resistance, concomitant with noteworthy enhancements in insulin sensitivity relative to control conditions [295].

In the DIANA-5 Study [NCT05019989], a trial of the MedDiet, physical activity, and BC recurrences, 2132 invasive BC patients surgically treated for stage I–III with no history of recurrences, metastases, or other cancers were randomized into control and intervention arms. The control group (CG) received general standard written recommendations for a healthy lifestyle without any active support, while the BC patients assigned to the intervention group (IG) received dietary reinforcement meetings over the course of one year, including cooking courses and gym and dance classes, with an emphasis on a comprehensive dietary and lifestyle change based on MedDiet and macrobiotic recipes and principles. Women in the intervention group were advised to engage in regular moderate-intensity physical activity, aiming for approximately 210 min per week. Although monthly structured physical activity sessions were initially suggested, participation was minimal. Consequently, after a few months, the structured physical activity program was restricted to small groups of volunteers. The dietary intervention trial of DIANA-5 failed to show a reduction in BC recurrence and metastasis. However, when the self-reported diet at year 1 in both the IG and CG was analyzed together, it revealed a protective association with a higher change in the Dietary Index [296].

The randomized clinical trial [NCT03314688], titled “Lifestyle, Exercise, and Nutrition Study Early After Diagnosis (LEANer),” aims to evaluate the effects of a dietary and physical activity intervention on women newly diagnosed with BC. The Healthy Eating Index-2015 was used as a measure of diet quality, while adherence to the physical activity guidelines was defined as ≥150 min/week of moderate- to vigorous-intensity physical activity or 75 min/week of vigorous-intensity physical activity and twice-weekly resistance training. Conducted by Yale University in collaboration with the National Cancer Institute, the trial involved 172 participants scheduled to receive neoadjuvant or adjuvant chemotherapy. The lifestyle interventions significantly improved various biomarkers in BC patients, including reduced insulin resistance, inflammation, and oxidative stress, as well as increased adiponectin levels and improved cholesterol and triglyceride profiles. Additionally, better pathological complete response rates were observed [297].

The active Phase 2 RCT [NCT04298086], titled “A Study of the Body’s Response to Exercise and a Plant-Based Diet in Overweight Postmenopausal Women With Breast Cancer,” aimed to investigate the combined effects of exercise and a plant-based diet on various health outcomes in overweight postmenopausal women with HR+ BC who are receiving aromatase inhibitor therapy. Females were randomly assigned into intervention and control groups. The intervention group followed a structured exercise program combined with a plant-based diet, while the control group maintained their usual diet and activity levels. Exercise treatment consisted of individualized walking delivered up to seven times weekly to achieve the patient-specific goal energy expenditure. Training sessions were performed on a treadmill under remote surveillance using a telemedicine approach established in the Exercise-Oncology Service. The results from the trial were presented in a poster during the 2024 American Society of Clinical Oncology (ASCO) Annual Meeting, highlighted several beneficial outcomes for the intervention group [298]. Women who adhered to the exercise regimen and plant-based diet experienced significant weight loss compared to the control group. This weight reduction was accompanied by improvements in body composition, including decreased body fat percentage and increased lean muscle mass [298].

Another completed trial, although its results are not yet published, is the CREATE trial [NCT03131024], which evaluates the impact of CR and exercise on mitigating anthracycline-induced toxicities in 56 women with early stage BC. Participants are divided into three groups: 50% CR for 48 h before each anthracycline treatment cycle, 30 min of vigorous-intensity aerobic exercise 24 h prior to chemotherapy, and a control group [299].

Ongoing trials, such as the Diet Restriction and Exercise-induced Adaptations in Metastatic BC (DREAM) trial [NCT03795493], aim to evaluate the therapeutic effect, adverse effects, and QoL of a short-term, 50% calorie-restricted KD combined with aerobic exercise during chemotherapy treatment for patients with metastatic BC [300].

Another ongoing trial, currently recruiting, studies the effects of combining both the MedDiet and exercise in reducing side effects in patients with stage I-IIIa BC receiving aromatase inhibitors [NCT03953157]. In this phase I/II trial, patients will be randomized into either a three-month anti-inflammatory MedDiet or bone-strengthening exercise arms to examine how this may alleviate medication side effects such as joint and bone pain and protectively influence bone mineral density, improve heart functioning, and reduce the risk of BC recurrence.

Other ongoing clinical trials, such as NCT04708860, NCT06123988 and NCT04174391 are still recruiting participants and aim to investigate the potential role of dietary regimens and physical activity in BC patients.

A preclinical study evaluated the antitumor effects of combining physical activity and calorie restriction in a BC mouse model [168]. This combined strategy effectively delayed tumor growth, reduced metastatic progression, and improved survival in the 4T1.2 mammary tumor model [168].

These data strongly support the need for large studies aimed at evaluating the combined effects of dietary interventions and exercise in BC patients.

## 9. Concluding Remarks

Our review provides compelling evidence that lifestyle interventions, including dietary regimens and physical activity may contribute to reducing the incidence and mortality of BC. Widely recognized as one of the healthiest diets, the MedDiet primarily consists of natural ingredients believed to mitigate oxidative stress and inflammation. As previously described, several physiological mechanisms underlie the overall protective effect of the MedDiet against BC. Rich in antioxidant substances, the MedDiet affects oxidative stress and DNA damage associated with cancer. Consumption of fruits and vegetables, abundant in dietary polyphenols, is commonly advocated for BC prevention and treatment. Olive oil, a staple of the MedDiet, is replete with flavonoids, phenolic compounds, triterpenes, and vitamin E, potentially contributing to its anti-inflammatory and antioxidant properties. Moreover, numerous scientific reports have suggested a protective effect of EVOO against BC. Emerging evidence indicates that the MedDiet exerts a beneficial effect on gut microbiota, thus playing a protective role in cancer. Several observational studies have indicated a potential protective effect of the MedDiet against BC. Unlike more restrictive dietary approaches, the MedDiet is considered one of the safest and most sustainable options for BC patients, providing essential nutrients without the risks of malnutrition or compromised immunity. Therefore, the MedDiet stands out as a balanced and effective dietary strategy, warranting further integration into BC care protocols to enhance patient well-being and treatment efficacy. Nevertheless, individualized dietary recommendations based on a patient’s unique medical history, nutritional needs, and preferences are essential for optimizing outcomes in BC management.

Fasting and MF have shown promise in the prevention and treatment of BC by targeting metabolic and immunological pathways. These dietary interventions were suggested to make chemotherapy more tolerable, reduce inflammation, and improve metabolic health. Preclinical studies suggest that fasting can inhibit tumor growth and increase the sensitivity of cancer cells to different types of treatments, including chemotherapy and ET that are commonly used in BC. Clinical studies, including two trials conducted by our team, enrolling 90 [NCT03595540] and 35 [NCT05748704] cancer patients, respectively, show the feasibility and safety of multiple fasting cycles in patients. Ongoing clinical trials are further investigating this approach, with the ultimate goal of defining whether fasting’s promising antitumor effects that were reported in animals hold true in patients, helping improve treatment outcomes and reduce side effects.

CR presents a promising avenue for BC prevention and treatment, supported by robust evidence from preclinical studies and emerging clinical data. By inducing favorable metabolic and hormonal changes, reducing inflammation, and enhancing immune responses, CR can create an environment less conducive to cancer development and progression. However, the implementation of CR must be approached with caution, particularly in cancer patients, to avoid adverse effects such as malnutrition and compromised immunity. Future research should focus on optimizing CR protocols and integrating them with existing cancer care guidelines to maximize patient outcomes while ensuring nutritional adequacy and overall health.

KDs have sparked interest in BC management due to their potential metabolic effects. Clinical studies suggest that a KD may inhibit cancer cell growth through mechanisms involving reduced glucose availability, increased insulin sensitivity, elevated levels of KBs, and potential weight loss, thus altering metabolic pathways. However, concerns include potential nutritional deficiencies, challenges in long-term adherence, and insufficient evidence regarding a KD’s long-term impact on BC outcomes and overall health. Therefore, while promising, KDs should be cautiously considered alongside conventional therapies and under medical supervision to balance potential benefits with nutritional and safety concerns.

Another significant dietary approach is the vegan or plant-based diet, which emphasizes plant-derived foods. These diets have been suggested to potentially reduce the risk of BC and improve outcomes by decreasing exposure to carcinogens, modulating gut microbiota, and lowering levels of IGF1, blood glucose, and cholesterol, as well as providing antioxidant and anti-inflammatory benefits. Adopting a vegan or plant-based diet can be a powerful tool in the fight against BC, offering potential benefits in prevention, improved survival rates, and enhanced QoL. Overall, it is important to approach dietary change thoughtfully and with professional guidance to ensure optimal health outcomes.

Structured physical activity is proposed to reduce the risk of developing BC and incidence of recurrences, prolong survival, and aid in recovery from anticancer therapies. Proposed mechanisms include changes in metabolic hormone concentrations (e.g., insulin, insulin-like growth factors, estrogens, androgens, and cortisol), modifications in the production and release of adipokines (e.g., adiponectin and leptin), decrease in low-grade inflammation by promoting anti-inflammatory cytokines (e.g., IL-10) and blunting the synthesis of pro-inflammatory cytokines (e.g., TNF-α and CRP), induction of epigenetic alterations to DNA (e.g., histone modifications, DNA methylations, expression of microRNAs, and changes in chromatin structure), enhancement of antioxidant defenses, and promotion of anti-tumor immune defenses (e.g., mitigation of immunological senescence, modulation of T cells, induction of phenotypic switch of macrophages, and activation and mobilization of NK cells). However, the relative contribution of each mechanism and their combined effects on cancer prevention/survival still need to be fully elucidated.

Collectively, while there is some evidence to suggest that the dietary regimens and physical activity may have potential benefits for BC patients, further research, including large-scale randomized controlled trials with long-term follow-up, is needed to elucidate the specific role of these lifestyle interventions in BC prevention and treatment.

## Figures and Tables

**Figure 1 nutrients-16-02262-f001:**
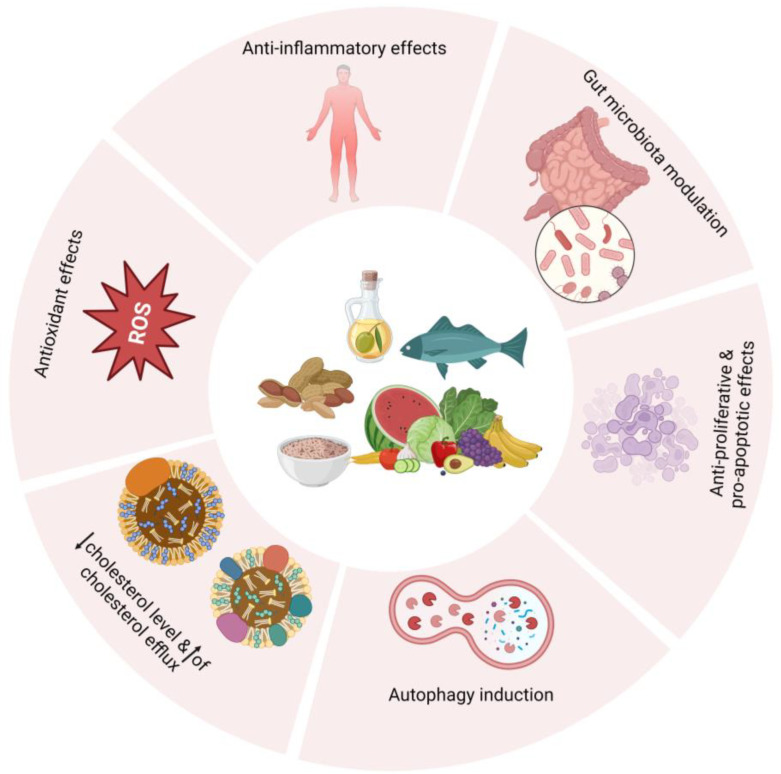
Summary of the potential mechanisms by which the MedDiet improves BC. ↑: increase in the levels/biological process, ↓: decrease in the levels/ biological process. (This figure was created with BioRender.com).

**Figure 2 nutrients-16-02262-f002:**
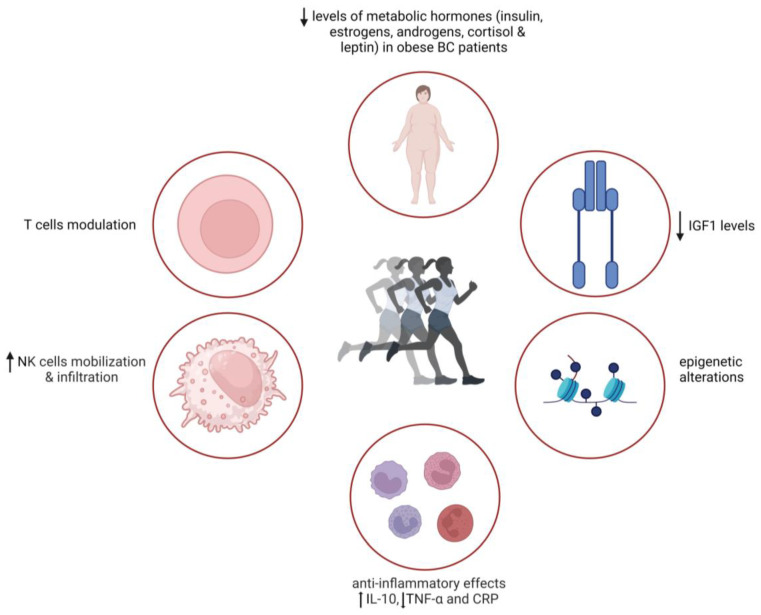
Overview of the potential mechanisms of physical activity in BC prevention and treatment. ↑: increase in the levels/biological process, ↓: decrease in the levels/biological process. This figure was created with BioRender.com.

**Table 1 nutrients-16-02262-t001:** Clinical evidence on the effects of the MedDiet in BC patients.

NCT Identifier	No. of Patients	Type of BC Patients	Notes/Results
NCT03953157	Recruiting (≈20)	Stage I-IIIa BC receiving AIs	The trial assesses if the MedDiet can alleviate AIs side effects, such as joint and bone pain, protect bone density, enhance heart function, and lower BC recurrence risk.
NCT04818996	25	Obese BC patients	This study aimed to evaluate the effects of the MedDiet on body composition, oxidative stress, and pro-inflammatory markers in overweight and obese women with BC. Although the study is completed, the results have not yet been published.
NCT04174391	Recruiting (≈766)	Female with primary pathologically confirmed invasive BC in stages I, II, or IIIA	This randomized, multicenter trial (LifeBreast) study will have two arms, with patients receiving (i) MedDiet + EVOO and (ii) low-fat diet to assess the efficacy of the applied nutritional intervention for the prevention of relapses among women with early BC.
NCT03314688	172	Newly diagnosed BC patients with stage I–III receiving chemotherapy	Healthy diet and exercise intervention did not impact RDI, but it did lead to improved pCR in patients.
ISRCTN35739639	4152	BC incidence, a prespecified secondary outcome, focused on women without a prior history of BC	Women in the ‘MedDiet + EVOO’ group saw a remarkable 62% decrease in invasive BC compared to the control group. Conversely, those in the ‘MedDiet + nuts’ group experienced a reduction in risk that did not reach statistical significance.
NCT05019989	2132	Surgically treated for stage I–III invasive BC patients	DIANA-5 randomized controlled trial did not find any effect of a Mediterranean macrobiotic diet on 5-year BC recurrence. However, high adherence to the diet was linked to a 41% lower risk of recurrence compared to controls.

BC: breast cancer, MedDiet: Mediterranean diet, AIs: aromatase inhibitors, MedDiet + EVOO: Mediterranean diet supplemented with extra-virgin olive oil, RDI: relative dose intensity, pCR: rate of pathologic complete response.

**Table 2 nutrients-16-02262-t002:** Clinical evidence on the effects of CR in BC patients.

NCT Identifier	No. of Patients	Type of BC Patients	Notes/Results
NCT02983279	49	NA	The aim of this trial is to evaluate whether 25% CR for 3–12 weeks prior to definitive cancer surgery affects tumor biology in breast, endometrial, or prostate cancers. The focus is on assessing the impact on biomarkers, particularly miR-21, an onco-miR known to influence cancer outcomes positively.
NCT01819233	38	Stage 0–I BC patients during surgery and radiation therapy	An ongoing feasibility trial in which patients were randomly assigned to either a CR group, which involved a specific diet plan reducing their caloric intake by 25% of their normal diet for 10 weeks, or a control group that continued with their regular diet.
NCT03131024	58	Early stage BC receiving anthracycline	BC patients were assigned into three groups: one group that undergoes 50% CR for 48 h before each anthracycline treatment cycle, another group that performs 30 min of vigorous-intensity aerobic exercise 24 h prior to chemotherapy, and a control group.
NCT04959474	80 estim.	DCIS or invasive BC patients	This phase II trial examines the impact of CR by 25% combined with sABR in BC patients. The trial aims to determine whether adding CR to radiation therapy can enhance local tumor control and reduce cancer spread.

BC: breast cancer, CR: calorie restriction, DCIS: ductal carcinoma in situ, estim.: estimated, sABR: stereotactic ablative radiation therapy.

**Table 3 nutrients-16-02262-t003:** Clinical evidence on the effects of KDs in BC patients.

NCT Identifier	No. of Patients	Type of BC Patients	Notes/Results
NCT02744079	11	ER+ or ER− BC patients following a breast mass biopsy	This randomized study compared the effects of various healthy diets on BC markers, specifically focusing on proliferation (Ki-67) and apoptosis (TUNEL) in surgical specimens.
NCT03535701	20	Stage IV BC AJCC v6 and v7	This non-randomized study evaluated adherence and compliance to a KD, along with changes in psychosocial measures, QoL, and physiological outcomes, including laboratory biomarker analysis (e.g., inflammation and tumor-related changes), to assess recurrence in patients treated with paclitaxel.
NCT02092753	150	Primary or recurrent BC	This non-randomized study analyzed whether a KD, compared to a LOGID or a SD, is feasible, safe, and tolerable and whether it can improve QoL and physical performance in BC patients during the rehabilitation phase.
NCT02516501	156	Breast carcinoma, rectum carcinoma, and head and neck cancer	This study assessed the feasibility of a KD during RCT and compared the effects of the KD to SD on BW, BIA phase angle, and quantities derived from BIA variables, QoL, toxicities, and blood parameters.

ER+: estrogen receptor-positive, ER−: estrogen receptor-negative, AJCC: American Joint Committee on Cancer, BC: breast cancer, KD: ketogenic diet, QoL: quality of life, LOGID: low glycemic and insulinemic diet, RCT: radio(chemo)therapy, SD: standard diet, BW: body weight, BIA: bioelectrical impedance analysis.

**Table 4 nutrients-16-02262-t004:** Clinical evidence on the effects of vegan or plant-based diets in BC patients.

NCT Identifier	No. of Patients	Type of BC Patients	Notes/Results
NCT03045289	32	MBC patients	This trial assessed the impact of a whole-food, plant-based diet on weight, cardiometabolic health, and hormonal outcomes. The study found that participants adhering to the diet experienced significant weight loss. Cardiometabolic health improved, evidenced by reductions in total and LDL-C, lower blood pressure, and better glycemic control. Hormonal changes included reduced levels of IGF1 and alterations in estrogen levels.
NCT03959618	200	BC patients receiving standard IV CT regimen	An ongoing clinical trial in France designed to determine the role of *Desmodium adscendens* [source of triterpene saponins, alkaloids, flavonoids, polyphenols, and tryptamine derivatives] as a neoadjuvant /adjuvant along with IV chemotherapy in BC patients.
NCT04298086	43	Overweight postmenopausal women with HR+ BC receiving AIs	Females in the intervention group (exercise with a plant-based diet) experienced notable weight loss compared to the control group. This weight reduction was accompanied by improvements in body composition, including decreased body fat percentage and increased lean muscle mass.
NCT03615599	app. 96,000	NA	The study examined the impact of plant-based diets on cancer risk among app. 96,000 participants, focusing on BC incidence within the cohort. Adherence to vegan diets was associated with a significantly lower risk of developing BC. Higher consumption of fruits, vegetables, whole grains, nuts, and legumes correlated with reduced BC risk, while higher intake of animal products and processed foods increased risk.

BC: breast cancer, MBC: metastatic BC, IGF1: Insulin-Like Growth Factor 1, LDL-C: low-density lipoprotein cholesterol, IV: intravenous, CT: chemotherapy, HR+: hormone receptor positive, AIs: aromatase inhibitors, app.: approximately.

**Table 5 nutrients-16-02262-t005:** Studies investigating the impact of physical activity in BC patients.

NCT Identifier	No. of Patients	Type of Physical Activity	Time	Results/Outcome
NCT01516190	49	Total of 40 m strength training + 180 m aerobic exercise per week in newly diagnosed patients before treatment.	29.3 days	Metabolic effects: decrease in leptin and IGF1 Immune effects: increase in inflammatory and immune modulatory activity in tumor tissue.
NCT01621659	80	Moderate/high intensity continuous and interval aerobic training, and moderate upper to lower extremity resistance training in patients receiving CT.	6–12 months	The outcome is to decrease chemotherapy-related cardiotoxicity and improve QoL.
NCT02522260	240	Strength/aerobic exercise, weights, bike, or treadmill twice a week in patients receiving CT.	16 weeks	No significant differences between groups. However, patients undergoing physical activity had a reduced hospitalization rate and a positive effect on thrombocytopenia.
NCT01515124	450	Two 60–90 min sessions/week of weight-lifting + 180 m of weekly aerobic exercise in patients diagnosed with BCRL.	52 weeks	The authors concluded that home-based training did not improve BCRL outcomes.
NCT02351479	23	Supervised 60 min of Hula dance twice/week and 15 min home-based exercise thrice/week.	6 months	Significant decrease in metabolic biomarkers (IGF1 and IGFBP-3) and multiple inflammatory cytokines such as IL-1β, IL-2, IL-4, IL-5, IL-6, IL-8, IL-10, GM-CSF, IFN- γ, and TNF-α, IL-10, GM-CSF.
NCT03658265	200	Shoulder isotonic exercise at different timelines in BC patients undergoing mastectomy.	12 weeks	To evaluate which physical activity routine in postoperative BC patients would prevent shoulder dysfunction.
NCT01147367	46	Total of 160 min moderate intensity walking and strength training with resistance bands.	3 months	Improvement of fatigue, sleep dysfunction, anxiety, and depression, while decrease in IL-10.
NCT00639210	487	Total of 60 min of supervised training once a week in patients who completed adjuvant CT or receiving ET.	1 year	Decrease in CT- and ET-related side effects and enhancement of QoL.
NCT03314688	172	Engage in 150 min of moderate to vigorous exercise or 75 min of vigorous exercise per week, along with resistance training twice a week.	2-year recruit., 5-year follow-up	Improved various biomarkers in BC patients. These included reduced insulin resistance, inflammation, and oxidative stress, as well as increased adiponectin levels and better cholesterol and triglyceride profiles. Besides, better pCR rates were observed.
NCT00640666	28	Total of 150 min of supervised moderate-intensity aerobic exercise per week and two weekly training sessions that gradually transition to home-based exercise.	3 months	Leptin and total adiponectin decrease, while the ratios of IL-6:IL-10, IL-8:IL-10, and TNF-α:IL-10 decrease.
NCT01331772	61	Weekly aerobic exercises in patients beginning adjuvant chemotherapy.	6 months	No statistically significant differences in circulating metabolic/inflammatory biomarkers levels were observed between exercise and control groups.
NCT02056067	121	Twice weekly supervised training sessions and 150 min of walking per week in BC survivors taking AIs.	12 weeks	Improved body composition, helped to decrease AIs-related side effects, and improved health outcomes in BC survivors undergoing exercise.
NCT01030887	26	Twice/week, moderate-intensity exercise in patients who had completed adjuvant CT.	8 weeks	No significant differences between the exercise and control groups regarding metabolic and inflammatory markers.
NCT00486525	200	Ninety min twice per week yoga classes in BC survivors.	12 weeks	High vitality, lower fatigue, and decreased IL-6, IL-1β, and TNF-α in patients receiving exercise.
NCT02408107	80	Thirty min of moderate-intensity counseling sessions in post-adjuvant therapy invasive patients.	6 months	Exercise reduced TC and LDL-C and improved QoL in the physical activity group.
NCT01140282	100	Supervised 60 min sessions thrice weekly and 30–45 min home-based sessions once weekly in BC survivors taking AIs.	16 weeks	Reductions in insulin resistance and pro-inflammatory and metabolic biomarkers (IGF1, leptin, adiponectin) as well as improvement in QoL and physical fitness in patients who received physical activity.
NCT02235051	50 estim.	Sessions of 30 min of supervised curves exercise 3 days per week.	16 weeks	This study will evaluate whether physical activity increases DNA repair capacity, reduces the inflammatory response, and modulates telomerase activity.
NCT02433067	89	Three training sessions on cycle-ergometer per week in HER2+ BC patients receiving trastuzumab.	3 months	Physical activity reduced trastuzumab-related cardiotoxicity.
NCT00405678	20	Sessions of 20 to 45 min of supervised aerobic cycle ergometry thrice per week in stage IIB–IIIC BC patients receiving 1st line NCT.	12 weeks	Modulated circulating pro-inflammatory cytokines as well as tumor gene expression pathways which have an important role in NF-κB signaling, Wnt/β-catenin, inflammation, oxidative phosphorylation, and cell migration.
NCT00115713	242	Three times per week of supervised aerobic exercise on a cycle, treadmill, or elliptical ergometer that lasted for 15–45 min.	18 weeks	There was a non-significant improvement in disease free survival in exercise arm.

IGF1: insulin-like growth factor 1, QoL: quality of life, BCRL: breast cancer–related lymphedema, CT: chemotherapy, ET: endocrine therapy, pCR: pathologic complete response, recruit.: recruitment, TC: total cholesterol, LDL-C: low-density lipoprotein cholesterol, AIs: aromatase inhibitors, estim.: estimated, NCT: neoadjuvant chemotherapy.

**Table 6 nutrients-16-02262-t006:** Preclinical studies of physical activity in BC mouse models.

BC Model	Mice Strain	Type of Physical Activity	Main Findings	Ref.
Subcutaneous injection of MCF-7 and MDA-MB-231 cells	NMRI-Foxn1nu	Running wheels	Exercise reduced tumor growth by 36% in MCF-7 tumors and by 66% in MDAMB-231 tumors, while also increased the regulation of the Hippo signaling pathway.	[146]
Triple-negative engineered mouse model	C3(1)SV40Tag	Treadmill running	Running resulted in 70% reduction in palpable tumors, decreased tumor volume, lowered plasma concentrations of MCP-1 and IL-6, and reduced spleen weight.	[147]
Subcutaneous injection of I3TC cells (murine mammary cancer line derived from the MMTV-PyMT Model)	FVB mice	Running wheels	The anticancer effects of exercise depend on CD8+ T-cells. Intense exercise can alter the intrinsic metabolism and antitumoral effector function of these cytotoxic T-cells.	[148]
Orthotopic implantation with EO771, 4TO7, or Transgenic mouse model	BALB/c, C57/Bl6, andC3(1)SV40Tagp16-luc	Treadmill exercise	Exercise treatment reduced EO771 tumor growth, increased C3(1)SV40Tagp16-luc tumor growth, and did not affect 4TO7 tumor growth that was dependent on Hif1-α expression.	[149]
4T1 mouse mammary tumor cells (allograft)	BALB/c	Aerobic exercise	Exercise increased tumor cytokines OSM and TNF-α. However, combining physical activity with SeNP reduced tumor volume and increased Th1 cytokines.	[150]
Orthotopic injection of MDA-MB-231 cells	Athymic	Voluntary wheel running	Exercise increased intratumoral vascularization in human BC xenografts.	[132]
4T1 cells	BALB/c	Aerobic exercise (treadmill)	Exercise reduced the levels of CCL2, CCL5, and CCR2 chemokines and the tumor volume of 4T1 allografts.	[151]
Subcutaneous injection of MCF-7 cells	BALB/c	Moderate intensity exercise (treadmill)	Exercise led to a reduction in inflammatory markers such as IL-6, IL-18, TNF-α, and CRP.	[152]
Orthotopic injection of EO771	C57BL/6	Voluntary wheel running	Physical activity alone reduced the number of CD8+ T cells. However, combining anti-PD-1 with exercise increased the percentage of CD8+ T cells.	[153]
Subcutaneous injection MDA-MB-231	Athymic	Progressive treadmill running	Exercise did not influence the anticancer efficacy of doxorubicin in BC xenografts.	[154]
Orthotopic injection of 4T1 cells	BALB/c	Steady low- and moderate-intensity exercise before and after tumor implementation	Exercise delayed BC growth and reduced tumor volume by inducing apoptosis and suppressing M2 macrophage polarization.	[155]
Transgenic mouse model	C3(1)SV40Tag	Running wheel	Exercise reduced tumor growth but did not inhibit tumor initiation.	[156]
Subcutaneous injection of 4T1 cells	BALB/c	Motorized wheelrunning	Mice subjected to 8 weeks of exercise before tumor injection exhibited a slower growth rate, enhanced survival, and improved antitumor immune response by increasing the CD8+/FoxP3+ ratio.	[157]
Transgenic mouse model	p53-deficient (p53+/−): MMTV-Wnt-1	Treadmill running	Exercise increased the rate of tumor development, decreased survival time, and raised the proportion of mice with multiple mammary carcinomas in a p53-deficient mouse model. However, there was no difference in IGF1 levels in the exercise group.	[136]
Subcutaneous injection of 4T1 cells	BALB/c	Treadmill running	Physical activity decelerated tumor progression, reduced tumor-induced MDSCs accumulation, and enhanced NK and CD8+ T cell activation. Exercise enhanced responses to dual anti-PD-1 therapy and radiotherapy.	[158]
Transgenic mouse model	FVB/N-MMTV-PyMT	Voluntary wheel running	Exercise counteracted muscle weakness, reduced intrinsic stress, and increased mitochondrial and antioxidant activities.	[159]
Orthotopic injection with EO771, EMT6, or MCa-M3C cells	C57BL/6, BALB/c and FVB	Treadmill running	Exercise delayed tumor growth, normalized tumor vessels, increased CD8+ T-cell infiltration, and synergistically enhanced effector CD8+ T cell activity when combined with ICB.	[160]
Transgenic mouse model	PyMT	Voluntary wheel running	Exercise decreased tumor sizes, with greater running distance correlating to smaller tumor size, and reduced CCL22 cytokine levels.	[161]
Orthotopic transplantation of E0771 cells	C57BL/6	Acute swimming after wheel running	Swimming for 45 min lowered glutamine levels, decreased tumor growth, and prevented muscle atrophy and weight loss in tumor-bearing mice.	[162]
Orthotopic injection with 4T1 cells	BALB/c	Treadmill running	The combination of physical activity and BBR slowed cancer progression by inducing immunomodulatory effects and apoptosis.	[163]
Orthotopic injection with 4T1 cells	BALB/c	Treadmill running	Exercise reduced tumor growth by decreasing mitochondrial OXPHOS.	[164]
Orthotopic engraftment with 4T1 cells	BALB/c	Voluntary wheel running	Physical activity reduced tumor growth, stimulated apoptosis, and induced vascular normalization and angiogenesis, leading to reduced hypoxia within tumors. When combined with CT, it delayed tumor growth.	[165]
Intraductal transplantation of EMT6 and 4T1 murine cells	BALB/c	Voluntary wheel running	Exercise per se reduced intratumoral hypoxia and improved the efficacy of doxorubicin while limiting its cardiotoxicity.	[166]
Orthotopic injection of EO771 cells expressing luciferase	C57BL/6J	Angled running wheel	Exercise suppressed tumor growth by enhancing the anti-tumor potential of mononuclear phagocytes in mammary tissue.	[167]
Orthotopic injection with 4T1 cells expressing luciferase	BALB/c	Running wheel	Exercise paired with 10% CR delayed tumor growth, decreased lung metastasis, and improved survival, surpassing single therapy. Moreover, the combined approach reduced expression of immunosuppressive and metastatic genes in the TME.	[168]
Orthotopic injection of EO771 cells	Hyperlipidemic ApoE^−/−^	Voluntary wheel running	Physical activity slowed primary and secondary tumor formation, decreased tumor hypoxia, and reduced metastasis.	[169]
Orthotopic engraftment with 4T1 cells	CB6F1	Voluntary running wheels	Rapamycin diet and running showed a significant increase in tumor burden.	[170]
Orthotopic injection of EO771 cells	C57BL/6	Treadmill running	Physical activity and vitamin D supplementation individually did not affect tumor growth. However, their combined treatment synergistically reduced weight gain in mice fed an HF diet.	[171]
Orthotopic injection of 4T1 cells	BALB/c	Treadmill running	Exercise and daidzein (phytoestrogens) together inhibited tumor growth through Fas/FasL-mediated apoptosis and regulated NK cell distribution via epinephrine and IL-6 upregulation.	[172]
Orthotopic injection of 4T1 cells	BALB/c	Swimming	Swimming reduced tumor growth by enhancing Th1 immune response, increasing Tbet and Nos2, elevating Th1-like cytokines, and decreasing Th2 profile. It also boosted IL-12 expression while reducing IL-4 and IL-10 expression.	[173]
7,12-dimethylbenz(a)anthracene BC carcinogen	BALB/c	Swimming	Eight weeks of swimming training in tumor-bearing BALB/c mice reduced splenic CD4+CD25+ Tregs and increased IFN-γ, TNF-α, and IL-12 cytokine expressions.	[174]
Orthotopic injection of MCa-M3C cells	FVB female syngeneic mice	Low, moderate, or high intensity exercise	Moderate intensity exercise reduced tumor growth and burden, increased tumor-infiltrating CD8+ T cells with enhanced function, and mobilized them into the bloodstream.	[158]

MCP-1: monocyte chemoattractant protein-1, IL: interleukine, Hif1-α: Hypoxia-inducible factor 1- α, TNF-α: tumor necrosis factor- α, OSM: oncostatin-M, SeNP: selenium nanoparticle, CCL: chemokine (C-C motif) ligand, CCR: C-C chemokine receptor, CRP: c-reactive protein, CT: chemotherapy, IGF1: insulin-like growth factor 1, MDSCs: myeloid-derived suppressor cells, PD-1: programmed cell death protein 1, NK: natural killer, ICB: immune checkpoint blockade, PyMT: polyoma middle T oncoprotein, BBR: berberine, OXPHOS: oxidative phosphorylation, CR: calorie restriction, TME: tumor microenvironment, HF: high-fat, FasL: Fas ligand, Tbet: T-box transcription factor TBX21, Nos2: nitric oxide synthase 2 (inducible), Th1: T-helper 1, Treg: T-regulatory.

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
