# Peer review of "Advances in Diet and Physical Activity in Breast Cancer Prevention and Treatment"

_nutrients, 2024, doi:10.3390/nu16142262_

Round 1

Reviewer 1 Report

Comments and Suggestions for Authors

General comments

- Based on the time and risk of the disease, the author of the review titled "Advances in diet and physical activity in breast cancer prevention and treatment" is significant.

- For patients with breast cancer and other cancers, the review is very helpful in improving their likelihood of survival after diagnosis and provides a strong opportunity to reduce both incidence and death. Appropriate but simple exercise along with a balanced diet and dietary restrictions are essential for preventing diseases like breast cancer.

Minor comments

The central point of the author's discussion is the significance of MedDiet. To better align with MedDiet, the title may be something like "Advances in MedDiet and physical activity in breast cancer prevention and treatment."

- Use of references published no more than five to ten years ago is advised due to the dynamic nature of health research. Several references that date back more than 15 or 20 years are not relevant to this review. For instance reference number 1, 2, 3, 11,13, 22, 39, 46, 87, 89, 90, 91, 107, 130, 135, 148, 159, and .....

- Create a mechanistic pathway summary graph that demonstrates how MedDiet, diet restriction, and exercise would help to regulate some significant signaling pathways and physiological situations in order to improve the review's readability and scientific interest.

Reviewer 2 Report

Comments and Suggestions for Authors

Not comprehensive and focused only/primarily on Med diet.

Should cover all other diets with the data in more detail 

Also, what about other cancers. So many reviews on the breast cancer topic already. Would be far more novel if focused on other cancers.

would be far more complete if a systematic review were performed.
